EMBO
Molecular Medicine

# Aggregation-resistant alpha-synuclein tetramers are reduced in the blood of Parkinson's patients

Laura de Boni [1,2,3], Amber Wallis[1], Aurelia Hays Watson[1], Alejandro Ruiz-Riquelme [4],
Louise-Ann Leyland[5], Thomas Bourinaris[6], Naomi Hannaway[5], Ullrich Wüllner[7,8], Oliver Peters[9,10],
Josef Priller [10,11,12,13], Björn H Falkenburger[14,15], Jens Wiltfang [16,17,18], Mathias Bähr[16,19,20], Inga Zerr[16,19],
Katharina Bürger[21,22], Robert Perneczky [21,23,24,25], Stefan Teipel[26,27], Matthias Löhle[26,28],
Wiebke Hermann[26,28], Björn-Hendrik Schott [16,29], Kathrin Brockmann[30,31], Annika Spottke[3,7],
Katrin Haustein[3], Peter Breuer[3], Henry Houlden[6], Rimona S Weil [5] & Tim Bartels [1]✉

## Abstract

Synucleinopathies such as Parkinson's disease (PD) are defined by the accumulation and aggregation of the α-synuclein protein in neurons, glia and other tissues. We have previously shown that destabilization of α-synuclein tetramers is associated with familial PD due to *SNCA* mutations and demonstrated brain-region specific alterations of α-synuclein multimers in sporadic PD patients following the classical Braak spreading theory. In this study, we assessed relative levels of disordered and higher-ordered multimeric forms of cytosolic α-synuclein in blood from familial PD with G51D mutations and sporadic PD patients. We used an adapted in vitro-cross-linking protocol for human EDTA-whole blood. The relative levels of higher-ordered α-synuclein tetramers were diminished in blood from familial PD and sporadic PD patients compared to controls. Interestingly, the relative amount of α-synuclein tetramers was already decreased in asymptomatic G51D carriers, supporting the hypothesis that α-synuclein multimer destabilization precedes the development of clinical PD. Our data, therefore suggest that measuring α-synuclein tetramers in blood may have potential as a facile biomarker assay for early detection and quantitative tracking of PD progression.

**Keywords** Alpha-synuclein; Blood; Human; Tetramer; Parkinson's disease
**Subject Categories** Biomarkers; Neuroscience

## Introduction

α-Synuclein is a protein found in neurons, where it plays a crucial role in the regulation of synaptic function. It is involved in dilation of the exocytotic fusion pore and subsequent neurotransmitter release or modulating glutamatergic receptor activity (Calabresi et al, 2023; Sharma and Burré, 2023; Logan et al, 2017). α-Synuclein promotes soluble n-ethylmaleimide-sensitive factor attachment protein receptor (SNARE) complex assembly by forming functionally active multimers upon membrane binding (Burré et al, 2014). Interestingly, wild-type α-synuclein and phosphorylated α-synuclein at the serine-129 site regulate neurotransmitter release (Parra-Rivas et al, 2023) upon neuronal activity (Ramalingam et al, 2023).

Thus, participation of α-synuclein in biological processes such as synaptic function requires conformational and posttranslational

[1]UK Dementia Research Institute, University College London, London W1T 7NF, UK. [2]Institute of Aerospace Medicine, German Aerospace Center (DLR), 51147 Cologne, Germany. [3]Department of Neurology, University Hospital Bonn, 53127 Bonn, Germany. [4]University of Santiago de Compostela, 15705 Santiago de Compostela, Galicia, Spain. [5]Dementia Research Center, Institute of Neurology, University College London, Queen Square, London WC1N 3BG, UK. [6]Department of Neuromuscular Diseases, UCL Queen Square Institute of Neurology, London WC1N 3BG, UK. [7]German Center for Neurodegenerative Diseases (DZNE), 53127 Bonn, Germany. [8]Departments of Neurology and Neurodegenerative Diseases, University Bonn, 53127 Bonn, Germany. [9]Institute of Psychiatry and Psychotherapy, Charité – Universitätsmedizin Berlin, corporate member of Freie Universität Berlin and Humboldt-Universität zu Berlin, 12203 Berlin, Germany. [10]German Center for Neurodegenerative Diseases (DZNE), 10117 Berlin, Germany. [11]Department of Psychiatry and Psychotherapy, Charité, Charitéplatz 1, 10117 Berlin, Germany. [12]University of Edinburgh and UK DRI, Edinburgh EH16 4SB, UK. [13]School of Medicine, Technical University of Munich, Department of Psychiatry and Psychotherapy, 81675 Munich, Germany. [14]German Center for Neurodegenerative Diseases (DZNE), 01307 Dresden, Germany. [15]Department of Neurology, University Hospital Carl Gustav Carus, Dresden University of Technology, 01307 Dresden, Germany. [16]German Center for Neurodegenerative Diseases (DZNE), 37075 Goettingen, Germany. [17]Department of Psychiatry and Psychotherapy, University Medical Center Goettingen, Georg August University, 37075 Göttingen, Germany. [18]Neurosciences and Signaling Group, Institute of Biomedicine (iBiMED), Department of Medical Sciences, University of Aveiro, 3810-193 Aveiro, Portugal. [19]Department of Neurology, University Medical Center, Georg August University, 37075 Göttingen, Germany. [20]Cluster of Excellence Nanoscale Microscopy and Molecular Physiology of the Brain (CNMPB), University Medical Center Göttingen, 37075 Göttingen, Germany. [21]German Center for Neurodegenerative Diseases (DZNE), 81377 Munich, Germany. [22]Institute for Stroke and Dementia Research, University Hospital, LMU Munich, 81377 Munich, Germany. [23]Department of Psychiatry and Psychotherapy, University Hospital, LMU Munich, 81377 Munich, Germany. [24]Munich Cluster for Systems Neurology (SyNergy) Munich, 81377 Munich, Germany. [25]Ageing Epidemiology Research Unit, School of Public Health, Imperial College London, London W6 8RP, UK. [26]German Center for Neurodegenerative Diseases (DZNE), Rostock-Greifswald, 17489 Rostock, Germany. [27]Department of Psychosomatic Medicine, Rostock University Medical Center, 18147 Rostock, Germany. [28]Department of Neurology, University of Rostock, 18057 Rostock, Germany. [29]Department of Psychiatry and Psychotherapy, University Hospital Göttingen, Georg August University, 37075 Göttingen, Germany. [30]German Center for Neurodegenerative Diseases (DZNE), 72076 Tübingen, Germany. [31]Hertie Institute for Clinical Brain Research, Department of Neurodegenerative Diseases, University of Tübingen, 72076 Tübingen, Germany. ✉E-mail: t.bartels@ucl.ac.uk

changes of the protein. Within a cell, α-synuclein exists as a soluble, disordered monomer, characterized by a lack of a well-defined secondary, quaternary or tertiary structure (Michell et al, 2005; Fauvet et al, 2012; Phillips et al, 2015). However, α-synuclein can form an amphipathic helix when binding to highly curved membranes (Das and Eliezer, 2019; Westphal and Chandra, 2013). Alongside the disordered monomers, α-synuclein can undergo additional conformational changes (three-dimensional arrangement of the subunits in a multi-subunit protein; quaternary structure) and form higher-ordered multimers, such as tetramers in various cells and tissues (Bartels et al, 2011; Wang et al, 2011; Boni et al, 2022; Burré et al, 2014; Dettmer et al, 2015b; Nuber and Selkoe, 2023; Nuber et al, 2021; Wang et al, 2014; Fernández and Lucas, 2018). It is presumed that α-synuclein monomers are prone to aggregation, whereas α-synuclein multimers are resistant to aggregation (Dettmer et al, 2015b; Boni et al, 2022). Based on current understanding, the assembly of α-synuclein into multimers is critical for maintaining a ratio between aggregation-prone monomers and aggregation-resistant tetramers. A change of this ratio with bias towards monomeric forms is potentially linked to the initiation of disease (Dettmer et al, 2015b). It has been postulated that the helical tetramer conformationally stabilizes α-synuclein diminishing the aggregation propensity under patho-logical conditions (Dettmer et al, 2015a) as the disordered α-synuclein monomers can form β-sheet-like oligomers, fibrils and intracellular inclusions known as Lewy bodies, Lewy neurites or glial cytoplasmic inclusions (Serpell et al, 2000; Alam et al, 2019; Vilar et al, 2008). These pathological aggregates and inclusions are thought to interfere with normal cellular function, leading to neuronal death and the clinical manifestations of Parkinson's disease (PD), dementia with Lewy bodies (DLB), or multiple system atrophy (MSA) (Alam et al, 2019). Oligomeric species are considered the most cytotoxic forms of α-synuclein (Fusco et al, 2017; Cremades et al, 2012; Alam et al, 2019).

The presented work centers on examining the ratio between disordered α-synuclein monomers and higher-ordered multimers as the protein's physiological configurations. The current work does not examine pathological α-synuclein forms such as β-sheet-like oligomers, fibrils, or Lewy bodies. Within this framework, we define multimers as any physiological assembly of synuclein monomers exceeding one.

In addition to nerve cells, α-synuclein can also be found in cerebrospinal fluid (CSF), saliva, the retina, blood, or skin (Chahine et al, 2020; Ganguly et al, 2021). Indeed, it should be noted that beyond its presence in neurons, α-synuclein is found in red blood cells being one of the 20 most abundant proteins (Bryk and Wiśniewski, 2017; Barbour et al, 2008). Thus, based on α-synuclein distribution, the synucleinopathies should be considered multi-systemic diseases involving different tissues and organs.

As a potential biomarker for neurodegenerative diseases, numerous attempts have been made to measure levels of total α-synuclein from the human brain or CSF. The detection process however carries several obstacles:

1. Access: CSF collection requires skilled staff and is relatively invasive for patients.
2. Lack of specificity: Since α-synuclein is an ubiquitous protein found in healthy individuals, the presence of α-synuclein in the brain or CSF is not specific to neurodegenerative diseases (Jakes

et al, 1994). In PD and other synucleinopathies, decreased levels of α-synuclein can be found in CSF as a result of the underlying synucleinopathy (Ganguly et al, 2021; Magalhães and Lashuel, 2022). In contrast, CSF levels could be increased due to synaptic damage or altered release of α-synuclein in the extracellular space depending on neuronal activity (Burré et al, 2018; Brás et al, 2022). In addition, especially CSF samples can exhibit elevated α-synuclein levels due to blood contamination. Overall, there is a great variability in total α-synuclein levels and an ongoing debate whether increased or decreased total α-synuclein levels reflect pathological processes (Magalhães and Lashuel, 2022). In this context, the formation of insoluble aggregates (oligomers or fibrils) is probably more relevant to the disease pathology.

3. Detection challenges: The unfolded monomeric form of α-synuclein is challenging to detect accurately and reproducibly due to its low abundance in CSF and its tendency to aggregate, leading to inconsistent results (Fayyad et al, 2019).
4. Dynamic nature: α-synuclein exists in different forms in the cytosol, at membranes (e.g., the presynaptic plasma membrane (Man et al, 2021)), including monomers, physiological multimers, toxic oligomers, and fibrils. The dynamic equilibrium between these forms complicates the interpretation of the data obtained from the analysis of total α-synuclein (Selkoe et al, 2014; Quinn et al, 2012).
5. Lack of correlation with disease progression: The levels of total α-synuclein do not necessarily correlate with the severity or progression of neurodegenerative diseases, limiting its utility as a reliable biomarker (Fayyad et al, 2019; Magalhães and Lashuel, 2022).

More specific measures for the detection of β-sheet-like α-synuclein and other pathological aggregates in CSF have remerged recently. Novel techniques to study the presence and amount of pathological protein aggregates, the seed amplification assays (SAA), comprising real-time quaking induced conversion (RT-QuiC) or protein misfolded cyclic amplification (PMCA) have recently shown great promise (Siderowf et al, 2023). These assays aim to replicate the pathological process observed in diseases, where β-sheet-like proteins, such as α-synuclein or tau, form aggregates and spread throughout the brain (Standke and Kraus, 2023). These techniques employ specific conditions and detection methods to monitor the aggregation process, usually involving the use of fluorescent dyes or antibodies that selectively bind to the aggregated forms of the protein (Standke and Kraus, 2023). SAA therefore might allow the biochemical diagnosis of PD providing information about different subclasses within a disease and can detect prodromal individuals (Siderowf et al, 2023).

While seed amplification assays can replicate the aggregation process of β-sheet-like, aggregated proteins, they may not fully capture the dynamic progression and complexity of neurodegen-erative diseases. The choice of seeds used in the assay can impact the results. Different aggregated protein strains or conformations may have distinct properties and propagate differently. It is crucial to carefully select and characterize the seeds to ensure reproduci-bility and relevance to the disease being studied. Additionally, the variability in seed preparations across studies can make direct comparisons challenging (Standke and Kraus, 2023). SAA have been used in various tissues, especially CSF (Yoo et al, 2022), but blood-based SAA are still challenging. Serum albumin is one of the

main interaction partners of reactant α-synuclein and hence the major complicating factor in quantification of SAAs in blood serum (Vaneyck et al, 2023; Bellomo et al, 2019). Potentially, interactions with serum components limit the configurational freedom of α-synuclein (Vaneyck et al, 2023). More important, lipoproteins which are abundant in blood form complexes with α-synuclein impairing seed amplification in SAA (Bellomo et al, 2023). So far, only immunoprecipitation-based real-time quaking-induced conversion (IP/RT-QuIC) has been successful to detect pathogenic α-synuclein seeds in serum (Okuzumi et al, 2023).

Research on non-aggregated, non-toxic, higher-ordered α-synuclein tetramers has provided new insights into the molecular mechanisms underlying PD and has opened up new possibilities for the development of assays, biomarkers and therapeutic strategies (Fanning et al, 2019; Imberdis et al, 2019; Nuber et al, 2022; Nuber et al, 2021). Some of our recent studies have suggested that the ratio of tetrameric:monomeric α-synuclein could serve as a more reliable biomarker, as it might better reflect the pathological changes occurring in the brain (Boni et al, 2022; Dettmer et al, 2015b). The conversion of α-synuclein tetramers to disordered monomers or other multimeric, non-toxic species is thought to contribute to disease initiation and the formation of pathological aggregates, such as Lewy bodies (Nuber et al, 2022; Nuber et al, 2021; Imberdis et al, 2019; Dettmer et al, 2013; Dettmer et al, 2015b; Wang et al, 2011). However, these non-aggregated, non-toxic, higher-ordered, aggregation-resistant tetramers have not yet been examined in blood from PD patients, or compared to healthy controls.

As monomers and tetramers of α-synuclein are well-characterized physiological species, we focused on the investigation of physiological, disordered, aggregation-prone monomeric α-synuclein and physiological, cytosolic, helically-folded, tetrameric α-synuclein in human blood and analyzed changes of the α-synuclein protein in well-characterized genetic, familial PD (fPD) forms i.e., G51D mutation in the α-synuclein gene (SNCA) and sporadic PD (sPD) from two independent cohorts.

# Results

## The ratio of α-synuclein tetramers and monomers can easily be assessed using cross-linking of blood tissue

To address the question of whether the ratio of α-synuclein tetramers:monomers is also disturbed in peripheral tissue in sporadic synucleinopathies, we adapted our established tetramer assay (using cross-linking and Western blot analysis of lysate (Boni et al, 2022)) to analyses of frozen human whole blood samples (Fig. 1). This protocol revealed a prominent cytosolic ~60 kDa and 14 kDa α-synuclein species (Figs. 2A and EV1A) besides other minor multimeric bands, as described previously for blood and red blood cells (Bartels et al, 2011). Immunoprecipitation (IP) and subsequent Mass spectrometry guided protein identification of the isolated 60 kDa band, in comparison to mock IP, only identified α-synuclein specifically associated with the multimeric protein band (Table EV1). The analysis of the undigested 60 kDa band via electrospray ionization (ESI) Mass spectrometry further revealed a 58 kDa complex of α-synuclein protein (theoretical molecular weight of tetrameric α-synuclein 58 kDa) indicating that the ~60 kDa band observed in Western blot does represent a physiological, homo-multimeric (tetrameric) assembly consisting of the α-synuclein protein only (Fig. EV1B and Table EV1).

For the tetramer:monomer analysis, EDTA-blood samples were freshly taken, aliquoted, and frozen within 1–2 h after sampling. Due to protein degradation, the Western blot signal intensities and consequently tetramer:monomer ratios dropped when blood samples remained at room temperature (RT) for more than 6 h probably due to protein degradation (Fig. EV2A,B). In contrast, Western blot signal intensities remained stable after multiple freeze thaw cycles (Fig. EV2C). Furthermore, we investigated whether any underlying signal from Hemoglobin (62 kDa) interfered with the 60 kDa α-synuclein tetramer signal diminishing the tetramer:monomer ratio. To this end, we choose 60 random samples and correlated the signal intensities of the remaining hemoglobin with signal intensities of the tetramer:monomer ratio at the lowest HemogloBind application (Blood:Hemoglobind 1:1, Fig. EV2D). We could not detect a correlation of either signal intensities after hemoglobin depletion (Fig. EV2D, $r = 0.2$, $p = 0.2$) indicating no interference between hemoglobin and α-synuclein tetramer detection. In addition, application of reducing agents and thereby removal of intact hemoglobin during preparation of samples before SDS-Page generally led to an improvement of the Western blot signals (Fig. EV3A).

### Characteristics of included patients and controls

See Table 1 for a summary of demographic and clinical characteristics in the genetic cohort (top), sPD cohort 1 (middle), and 2 (bottom). The UK cohort is similar to patients described in previous work (Leyland et al, 2020). Two different cohorts of sporadic PD were included: one from the UK (cohort 1) and one from Germany (cohort 2, Table 1). For cohort 1, 82 participants were involved in the study that included 65 patients with sPD (33 female), and 17 controls (9 females). For cohort 2, 147 participants were involved in the study, including 64 patients with PD (13 female); and 83 controls (40 females).

## The ratio of cytosolic α-synuclein tetramer:monomer is disturbed in PD patients with G51D mutations and G51D carriers

We assessed the ratio of cytosolic α-synuclein tetramer:monomer in a case study of patients with rare G51D mutations (Table 1) as other multimeric bands were not as apparent in all samples using different cross-linkers at a variety of concentrations (Figs. 2A,B and EV1A). α-Synuclein tetramer:monomer ratios were reduced in G51D carriers ($n = 2$) compared to controls ($n = 3$, Fig. 2A,B). The α-synuclein tetramer:monomer ratio was further diminished in a G51D patient who had already developed PD clinically (REM sleep disorder, dementia, rigor, akinesia, dyskinesia, hallucinations) compared to non-symptomatic G51D carriers and controls (Fig. 2A,B). In contrast, ratios of the control protein DJ1 (dimer/monomer) were not significantly different between controls, G51D carriers and the PD G51D patient (Fig. EV3B).

## The ratio of α-synuclein tetramer:monomer is disturbed in sporadic PD patients

We next assessed the ratio of α-synuclein tetramer:monomer ratio in sPD compared to controls in the two sporadic PD cohorts. All samples were analyzed in technical duplicates at least. Cohort 1 was

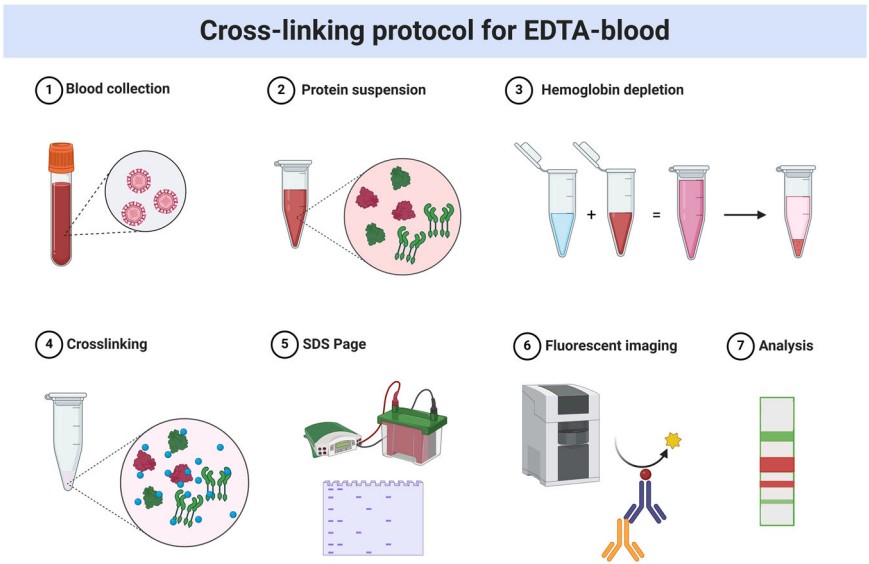

**Figure 1. Cross-linking protocol for EDTA-blood.**

Protocol and workflow. The procedure is described in the Methods section. Created with BioRender.com. Agreement number: JT25I9XJNL.

analyzed using the cross-linker DSG at a final concentration of 1.34 mM, cohort 2 was analyzed using the cross-linker GA at a final concentration of 0.0067%.

α-Synuclein tetramer:monomer ratios were significantly diminished comparing sPD patients to controls in both cohorts (cohort 1 $p = 0.02$, U = 348, cohort 2 $p < 0.001$, U = 1913, Fig. 3A,B). The ratio of the control protein DJ1 was not significantly different between both groups (cohort 1 $p = 0.2$, U = 449, cohort 2 $p = 0.8$, U = 2604), neither did the amount of total α-synuclein levels in the soluble or membrane-associated (Triton-X soluble) fractions as measured by ELISA (Fig. EV3C). Triton X-insoluble α-synuclein was not detectable via ELISA in blood.

Using Pearson's correlation analysis, we detected a significant negative correlation comparing α-synuclein tetramer:monomer ratios to disease duration ($r = -0.4$, $R^2 = 0.1$, $p = 0.004$, Fig. 4A) and the summary cognitive score ($r = -0.2$, $R^2 = 0.06$, $p = 0.04$, Fig. EV4E) in sPD patients from cohort 1. There was no significant correlation comparing α-synuclein tetramer:monomer ratios in sPD and controls to age, gender, UPDRS motor score, MMSE, MoCA or hallucinator status (Figs. 4C and EV4A–D,F). Correlation analysis of cohort 2 did not reveal a significant correlation comparing α-synuclein tetramer:monomer ratios to disease duration in sPD patients ($r = -0.07$, $R^2 = 0.005$, $p < 0.6$, Fig. 4B) and all other variables in sPD and controls (age, gender, MMSE, MoCA, UPDRS motor score, Figs. 4D and EV5A–D).

## Discussion

The α-synuclein tetramer:monomer ratio in blood was lower in patients with sPD, G51D carriers and G51D patients with clinical PD. In one sPD cohort, a relationship between disease duration, the summary cognitive score and blood-based α-synuclein tetramer:monomer ratios was apparent, which was not replicated in a second other cohort.

G51D is one of several point mutations of the *SNCA* gene besides A30P, E46K, H50Q, A53T, A53E, A53V, and A30G, that are associated with an autosomal inheritance of PD (Si et al, 2017; Liu et al, 2021; Nussbaum, 2018). Aggregation-resistant helical α-synuclein tetramers are partially destabilized by fPD causing α-synuclein missense mutations such as G51D (Dettmer et al, 2015b). In engineered neural cells, the insertion of G51D-analogous mutations (V40D + G51D + V66D) exacerbates the reduction of the α-synuclein tetramer:monomer ratio, increases cellular toxicity and reduces cell growth (Tripathi et al, 2022). Thus, it is important to note that while the G51D mutation is rare, its discovery and study have contributed valuable insights into the role of α-synuclein (encoded by the *SNCA* gene) in PD and other synucleinopathies. Despite our small samples size of G51D patients, representing a case study, studying the genetic form of PD associated with mutations in the *SNCA* gene (such as the G51D mutation), is valuable for several reasons, even in the context of sporadic PD. Understanding how these mutations lead to PD symptoms can shed light on the broader mechanisms involved in the sporadic form of the disease. In this context, it is interesting that we found differences in relative tetramer:monomer levels present in asymptomatic G51D carriers before symptom onset, suggesting a quantitative correlation with disease progression.

Moreover, clinical presentations and the distribution of α-synuclein pathology points towards PD as a heterogenous systemic disease (Wüllner et al, 2023; Simonsen et al, 2016). Consequently, blood may exhibit a fingerprint of the disease and clearly would be the tissue of choice for biomarker analysis. The major source of α-synuclein in blood is in erythrocytes (Barbour et al, 2008), where physiological α-synuclein is essential for exocytosis/endocytosis, apoptosis, autophagy, maturation and differentiation of hematopoietic cells (Pei and Maitta, 2019). So far, levels of physiological and pathological, aggregated α-synuclein have been examined in plasma, serum, erythrocytes, and peripheral

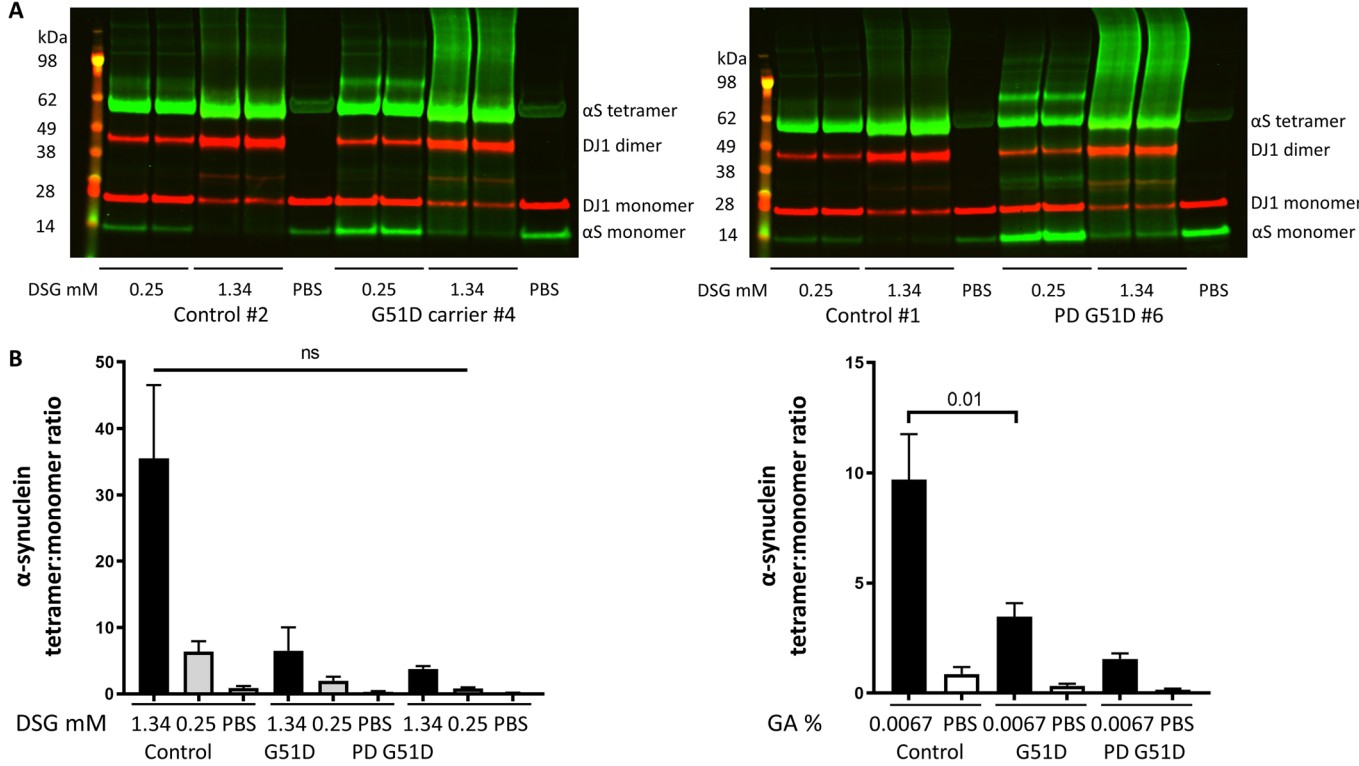

**Figure 2. Western blot analysis of blood lysate from G51D carriers and one PD G51D patient compared to controls.**

(A) Representative pictures of Western blot analyses from controls, G51D carriers and PD patients with G51D mutations using the cross-linker DSG. All samples have been analyzed in technical duplicates. Information on clinical characteristics is provided in Table 1. (B) Quantitation of Western blot signal shows lower α-synuclein tetramer:monomer ratios in G51D carriers ($n = 2$) and the G51D PD patient ($n = 1$) compared to controls ($n = 3$) using 2 different concentrations of the cross-linker DSG (left panel, 0.25 mM and 1.43 mM). Using a second cross-linker, GA, α-synuclein tetramer:monomer ratios were significantly reduced in G51D carriers (right panel, $p = 0.01$) compared to controls. The α-synuclein tetramer:monomer ratio was further diminished in one G51D patient who already clinically developed PD compared to non-symptomatic G51D carriers and controls. All groups were compared using Mann–Whitney-U test and are displayed as mean ± s.e.m. GA = Glutaraldehyde, DSG = Disuccinimidyl glutarate. Samples for Fig. 2 (depleting hemoglobin, cross-linking, gels, blots) were processed in parallel on different blots due to the samples size. No loading controls were run on the western blot as the full volume of each processed sample containing 20 μg of total protein was loaded into each gel pocket. Source data are available online for this figure.

blood mononuclear cells in PD, albeit with inconsistent results (Ganguly et al, 2021; Simonsen et al, 2016). Similar to CSF, plasma levels of total α-synuclein alone cannot robustly distinguish PD patients from healthy controls. As disordered, monomeric α-synuclein is prone to misfolding and self-association into high molecular weight, toxic species (Dobson, 2003), recent studies investigated oligomeric α-synuclein in blood. Some studies reported elevated oligomeric α-synuclein in erythrocytes and in blood from PD patients compared to healthy controls (Liu et al, 2022; Yu et al, 2022; El-Agnaf et al, 2006). However, other studies showed that oligomeric α-synuclein was not significantly increased in plasma or red blood cells (Fayyad et al, 2019). The aforementioned SAA is limited by the detection of fibrillar aggregates, and modified SAA to detect pathological α-synuclein conformations have been developed only recently (Okuzumi et al, 2023). Therefore, a blood-based biomarker with a quantifiable correlation with disease progression would further facilitate diagnosis, prognosis and therapy development through more sensitive and facile target engagement studies. Thus, we propose an additional approach by investigating the relative levels of physiological aggregation-resistant α-synuclein tetramers and other multimers compared with aggregation-prone

α-synuclein monomers in blood. Our results suggest that the relative abundance of physiological monomers compared with aggregation-resistant tetramers may form the basis of a potential marker of early disease in PD.

However, the exact biological role of α-synuclein tetramers, other multimers, and α-synuclein monomers in blood, especially erythrocytes, is not known yet. In mature erythrocytes, α-synuclein exerts structural functions, by tethering cytoskeletal proteins and the cytoplasmic membrane, as well as regulating metabolic activities such as iron homeostasis or mediating assembly of SNARE complexes in platelets and leukocytes (Barba et al, 2022; Pei and Maitta, 2019). Furthermore, α-synuclein monomers play an important role during hematopoiesis (Pei and Maitta, 2019; Barba et al, 2022). Notably, α-synuclein monomers and high molecular bands of α-synuclein are detected in the erythroid proliferative stage, while α-synuclein monomers are only detected in the erythroblast stage (Pei and Maitta, 2019). In addition, α-synuclein monomers are found in the nucleus, the cytoplasm or the plasma membrane depending upon the erythrocyte differentiation stage (Pei and Maitta, 2019). It is proposed that α-synuclein monomers tether other proteins to the membrane while being attached to the

**Table 1. Summary of demographic and clinical characteristics of genetic and sporadic PD cohorts.**

| Variable | Classification | Age [years] | Gender | Disease duration [years] |
|---|---|---|---|---|
| *Genetic cohort PD (UK)* | | | | |
| Case #1 | Control | 57 | Male | NA |
| Case #2 | Control | 53 | Female | NA |
| Case #3 | Control | 60 | Female | NA |
| Case #4 | G51D Carrier | 57 | Female | NA |
| Case #5 | G51D Carrier | 53 | Male | NA |
| Case #6 | PD patient, G51D mutation | 56 | Female | 16 |
| **Variable** | **PD** | **Control** | **Statistic** | **P** |
| *Cohort 1 sPD (UK)* | | | | |
| Number | 65 | 17 | NA | NA |
| Gender (F:M) | 1:1 | 1:1.1 | OR | 0.9 |
| Age [years] | 65 (2.3) | 60.2 (17.9) | U | 0.7 |
| Disease duration [years][a] | 5.3 (2.5) | NA | NA | NA |
| MDS-UPDRS[b] | 22.7 (10.6) | 4.5 (3.6) | U | <0.0001 |
| MMSE[b] | 29.1 (1.2) | 29.5 (0.5) | U | 0.3 |
| MoCA[c] | 27.9 (2.4) | 27.2 (2.6) | U | 0.4 |
| Summary cognitive score[d] | −0.3 (0.9) | −0.2 (0.8) | U | 0.6 |
| Hallucination severity[e] | 0.4 (0.9) | 0 | U | 0.2 |
| *Cohort 2 sPD (Germany)* | | | | |
| Number | 64 | 83 | NA | NA |
| Gender (F:M) | 3.9:1 | 1.1:1 | OR | 0.0007 |
| Age [years][f] | 66.3 (9.7) | 60.9 (14.1) | U | 0.03 |
| Disease duration [years][g] | 7.2 (5.4) | NA | NA | NA |
| MDS-UPDRS[h] | 30.7 (14.8) | NA | NA | NA |
| MMSE[i] | 28.1 (1.3) | 29.3 (0.9) | U | 0.01 |
| MoCA[j] | 26.5 (2.8) | 26.6 (2.2) | U | 0.8 |

All values are mean (SD) unless indicated.
*(s)PD* (sporadic) Parkinson's disease, *F* female, *M* male, *MDS-UPDRS* Movement Disorders Society Unified Parkinson's Disease Rating Scale, *MMSE* mini mental status examination, *MoCA* Montreal Cognitive Assessment, summary cognitive score =mean of the z-scores of the MoCA plus one task per cognitive domain, *hallucinator scale* University of Miami Parkinson's Disease Hallucination Questionnaire (UM-PDHQ), *OR* odds ratio, *U* Mann–Whitney-U-Test.
[a]Data only available for 64 sPD.
[b]Data only available for 64 sPD and 13 controls.
[c]Data only available for 13 controls.
[d]Data only available for 63 sPD and 13 controls.
[e]Data only available for 64 sPD and 13 controls.
[f]Data only available for 64 sPD.
[g]Data only available for 63 sPD.
[h]Data only available for 43 sPD and no controls.
[i]Data only available for 7 sPD and 45 controls.
[j]Data only available for 41 sPD and 21 controls.

cell membrane by its N-terminus thus increasing membrane mechanical strength and regulating membrane fluidity (Pei and Maitta, 2019). Over 99% of blood α-synuclein is present in erythrocytes (Barbour et al, 2008). Mature erythrocytes lack a nucleus and most organelles, therefore they cannot express α-synuclein independently. One possible theory is that α-synuclein enters the blood from the CSF and may be taken up from the plasma into erythrocytes (Yang et al, 2020), another possibility is that it represents leftover protein during erythrocyte maturation. Notably, α-synuclein in erythrocytes can form heterocomplexes with Aβ and tau (Daniele et al, 2021; Zhang et al, 2022; Yang et al,

2020), suggesting a possible synergy between α-synuclein and other proteins. Together with seeding assays, amplifying pathological, β-sheet like, possible toxic α-synuclein species such as oligomers or fibrils, assays on physiological tetramers or other multimers could complement the clinical diagnostic procedure in PD.

## Limitations

Our approach for analyzing α-synuclein tetramers and monomers is based on cross-linking and subsequent SDS-Page and Western blotting of samples. The technical approach is not feasible for

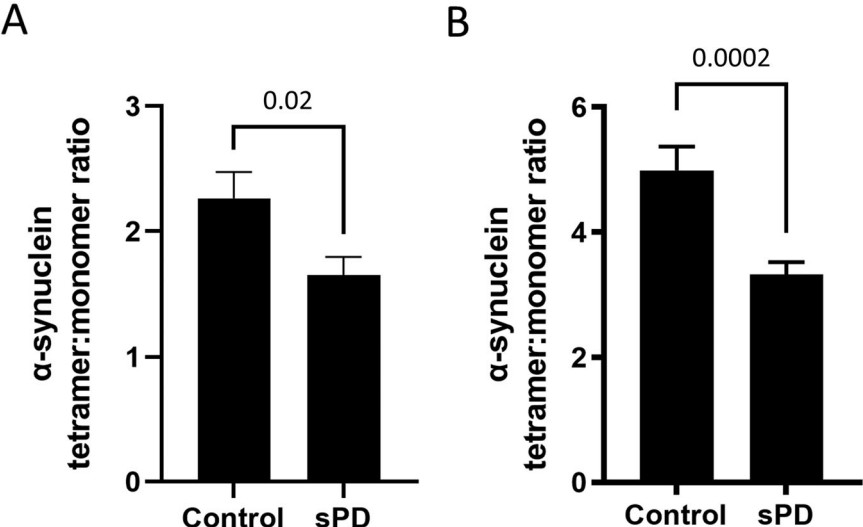

**Figure 3.  α-Synuclein tetramer:monomer ratios are diminished in sPD patients compared to controls.**

(A) Samples from cohort 1. α-Synuclein tetramer:monomer ratios are significantly reduced ($p = 0.02$) in sPD patients ($n = 65$) compared to controls ($n = 17$) upon cross-linking with DSG at a final concentration of 1.43 mM. Both groups were compared using Mann–Whitney-U test and are displayed as mean ± s.e.m. sPD = sporadic Parkinson's disease. All samples (depleting hemoglobin, cross-linking, gels, blots) were processed on different blots due to the samples size at least in technical duplicates. No loading controls were run on the western blot as the full volume of each processed sample containing 20 µg of total protein was loaded into each gel pocket. (B) Samples from cohort 2. α-Synuclein tetramer:monomer ratios are significantly reduced in sPD patients ($n = 64$) compared to controls ($n = 83$) upon cross-linking with GA at a final concentration of 0.0067% ($p = 0.0002$). Both groups were compared using Mann–Whitney-U test and are displayed as mean ± s.e.m. sPD = sporadic Parkinson's disease. All samples (depleting hemoglobin, cross-linking, gels, blots) were processed on different blots due to the samples size at least in technical duplicates. No loading controls were run on the western blot as the full volume of each processed sample containing 20 µg of total protein was loaded into each gel pocket. Source data are available online for this figure.

detecting precise, absolute levels of either tetramers or monomers. Thus, the analysis is limited on the internal ratio for tetramers:-monomers of each samples diminishing intra-individual variation. However, the approach allowed us to compare relative amounts between patients and controls. Furthermore, our approach here is time-consuming and laborious; signal detection can be impaired by improper blotting, smearing due to cross-linking and processing of the samples, as evidenced by some studies failing to demonstrate α-synuclein multimers due to technical reasons (Fauvet et al, 2012; Araki et al, 2016) while others independently succeeded demonstrating the presence of α-synuclein multimers (Kim et al, 2018; Abdullah et al, 2017; Iljina et al, 2016; Burré et al, 2014; Gould et al, 2014; Wang et al, 2014; Trexler and Rhoades, 2012; Westphal and Chandra, 2013; Wang et al, 2011). In addition, we only analyzed tetrameric, not any other multimeric bands as they were not present in all samples. Therefore, an improvement of the detection method, e.g., by using an α-synuclein-specific ELISA with primary antibodies binding directly to α-synuclein tetramers and other multimers would be beneficial. In addition, our data provides information on the homo-multimeric nature of the α-synuclein tetramer, however, transient protein interaction of the α-synuclein tetramer and other multimers should still be considered. Such an assay could also be applied to plasma samples providing a better fingerprint of the altered α-synuclein tetramer:monomer equilibrium in plasma (CSF-derived α-synuclein signature) vs. erythrocytes (peripheral α-synuclein signature).

While we detected a correlation of the α-synuclein tetramer:-monomer ratio with disease duration in cohort 1, we failed to detect a significant correlation in cohort 2 in sPD patients, possibly due to a higher variability in disease duration. We were also not able to show any significant correlation with patient age or other clinical assessments. Our results from cohort 2 indicate, that some sPD patients with longer disease duration exhibited relatively high tetramer:monomer ratios. It is unknown, whether individuals are born with certain multimer levels or whether tetramer:monomer or other multimer:monomer ratios change upon aging. The absence of a correlation of tetramer:monomer ratio with the age of the patients seem to imply that a pathogenic event is needed for α-synuclein destabilization to occur. In addition, we did not include patients with mild cognitive impairment or dementia in our cohorts, and did not observe significant changes in the MMSE or MoCA analysis comparing sPD and controls apart from the summary cognitive score which was available for cohort 1. Further analyses have to be performed on cognitive decline, as brain tissue analysis have already demonstrated a negative correlation between α-synuclein multimer:monomer ratio in sPD and DLB Braak 6 patients with dementia (Boni et al, 2022). So far, our data indicate, that a decrease in the tetramer:monomer ratio is causally linked to a pathological driver of disease progression, but not a higher susceptibility to develop disease due to general risk factors.

Overall, the easy accessibility of blood makes it an important body fluid as a biomarker to monitor levels of disease for diagnostic or prognostic evaluation (Ganguly et al, 2021). Improved assays performed on larger cohorts for the detection of α-synuclein tetramers and other multimers could determine whether α-synuclein multimers serve as a diagnostic or prognostic biomarker with predictive capability.

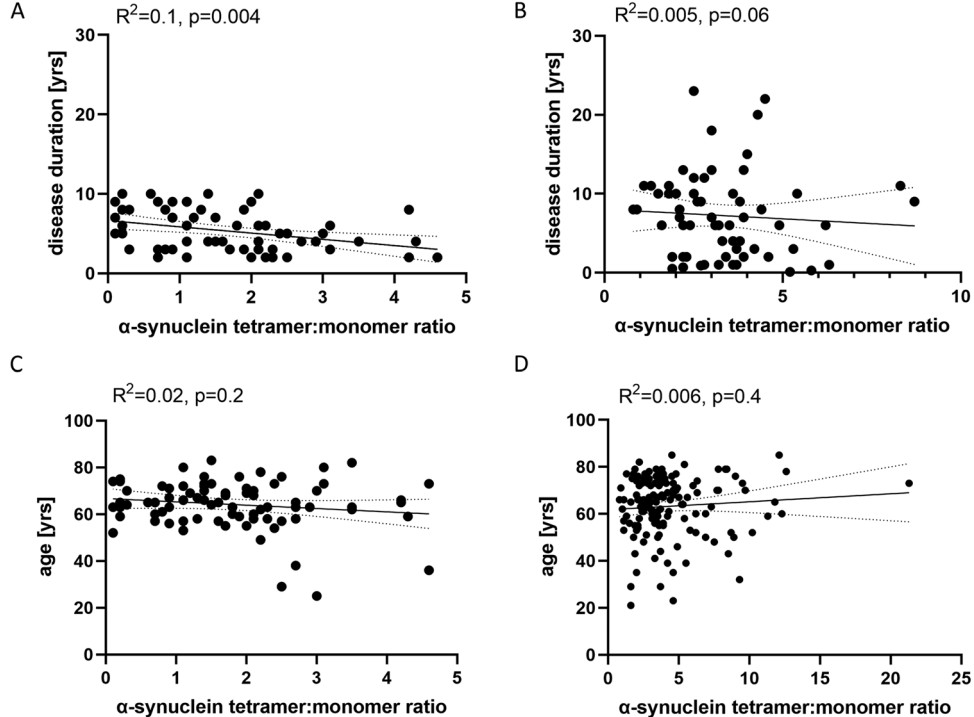

**Figure 4. α-Synuclein tetramer:monomer ratios are inversely correlated with disease duration but show no relationship with age.**

(A, B) Pearson correlation shows a negative correlation of α-synuclein tetramer:monomer ratios and disease duration for cohort 1 (sPD $n = 64$) (A) but not cohort 2 (sPD $n = 63$) (B) in sPD. (C, D) Pearson correlation shows no significant correlation of α-synuclein tetramer:monomer ratios and age for either cohort 1 (sPD $n = 65$, controls $n = 17$) (C) or cohort 2 (sPD $n = 64$, controls $n = 83$) (D) in sPD and controls. Yrs = years. Lines show 95% confidence bands of the best fit linear regression line. Source data are available online for this figure.

## Summary

We show that the ratio of α-synuclein tetramers:monomers is disturbed in blood from sPD and fPD patients; significant changes can be detected in single pre-symptomatic carriers of G51D mutations. Our findings suggest that α-synuclein tetramers and possibly other multimers represent a promising avenue of research for biomarker development in PD and related disorders. Further studies in larger patient cohorts are needed to validate the utility of physiological multimers and the multimer:monomer ratio as a valid, sensitive, and specific biomarker.

## Methods

### Patient recruitment and characterization

#### UK cohort

Patients with sPD were recruited to the Institute of Neurology at University College London (UCL) between October 2017 and November 2018. Inclusion criteria were a diagnosis of PD (using Queen Square Brain Bank Criteria), within 10 years' of diagnosis, aged 49–82 at the time of testing. Exclusion criteria were confounding neurological or psychiatric disorders, severe head injury, a diagnosis of dementia (mini mental state examination

(MMSE) score less than 25). G51D carriers and patients were recruited at the Department of Neuromuscular Diseases, UCL Queen Square Institute of Neurology.

Controls unaffected by neurological or psychiatric disease were recruited from family members, spouses, local volunteers, and university databases. All participants gave written informed consent and the study was approved by the Queen Square research Ethics Committee (ref. number 15.LO.0476). The experimental use of human blood samples was approved under the protocol RA063974/1.

#### Clinical evaluation

G51D and sPD patients were assessed whilst on their usual medications. Motor severity was assessed using the Movement Disorders Society Unified PD Rating Scale, part three (MDS-UPDRS-III) (Goetz et al, 2008). Cognitive severity was assessed using global measures of the MMSE and the Montreal Cognitive Examination (MoCA); a summary cognitive score was calculated for each participant as the mean of the z-scores of the MoCA plus one task per cognitive domain (see Hannaway et al (2023)). Presence of hallucinations was established using the MDS-UPDRS part one, with hallucination severity measured using the University of Miami Parkinson's Disease Hallucination Questionnaire (UM-PDHQ) (Papapetropoulos et al, 2008). Clinical information including disease duration was obtained through a semi-structured interview.

### German cohort

Clinical data and biomaterial samples were provided by the study group DESCRIBE and DANCER of the Biobank at the Clinical Research of the German Center for Neurodegenerative Diseases (DZNE). Exclusion criteria were confounding neurological or psychiatric disorders. The study was approved by the Ethics Committee of the Medical Faculty of Bonn (ref. number 075/20).

### Clinical evaluation

For the DESCRIBE study, test results obtained within the framework of routine medical care are recorded. These include medical history, medications taken, clinical neurological examination findings and the results of cognitive and motor skills assessments (see below). Motor severity was assessed using the MDS-UPDRS. Cognitive severity was assessed using the MMSE and MoCA. Clinical information including disease duration was obtained through a semi-structured interview.

All experiments were conformed to the principles set out in the WMA Declaration of Helsinki and the Department of Health and Human Services Belmont Report.

See Table 1 for a summary of all cohorts.

## Blood samples

For a schematic of the overall cross-linking process on blood, see Fig. 1.

Fresh frozen human EDTA-whole blood was thawed on ice and mixed 1:1 with 1x PBS buffer/1x phosphatase inhibitor/1x protease inhibitor (PBS/PI, Sigma, Thermo Fisher Scientific). Blood suspensions were lysed by sonication (Q800R3 Sonicator sonication of amplitude 20% for 15 s at 4 °C, Fisher Scientific Model 705 Sonic Dismembrator, sonication of amplitude 5% for 15 s at 4 °C or Bandelin Sonopuls, HD2070, SH70G, type MS72 10% for 15 s at 4 °C). Samples were centrifuged at full speed using a table microcentrifuge (Thermo Fisher, Eppendorf) for 30 min at 4 °C. Supernatant and pellet were stored independently at −80 °C.

## Hemoglobin depletion

Supernatant mixed with HemogloBind (Biotech Support Group) at a ratio of 1:2 and rotated at RT for 10 min. The sample was then centrifuged for 2 min at 14,000 rcf at 4 °C. The supernatant was transferred to a new tube.

## Cross-linking of blood lysate

The total protein amount of the hemoglobin-depleted blood samples (after lysis through sonication, see above) was measured with the Pierce BCA protein assay kit (Thermo Fisher Scientific) according to the according to the manufacturer's instructions. 20 μg of total protein from the lysate was added up to 10 μl total volume with PBS/PI (Sigma, Thermo Fisher Scientific). For the cross-linking of lysate, Glutaraldehyde (GA) and Disuccinimidyl glutarate (DSG) were used to test the effect of different cross-linkers. Glutaraldehyde solution (TAAB Laboratories) was added to the lysate at a final concentration of 0.0067%. The solution was incubated for 15 min at 37 °C shaking. DSG (Thermo Fisher Scientific) was added to the lysate at a final concentration of 0.25 or 1.43 mM. The solution was incubated for 30 min at 37 °C shaking.

Cross-linking reaction was quenched with 1 m Tris-HCl (Sigma). All samples have been analyzed at least in technical duplicates.

## SDS-Page, immunoblotting, and imaging

Samples were boiled at 70 °C in 5 μl (for GA cross-linking) or 10 μl (for DSG cross-linking DSG) 4 × NuPage LDS sample buffer (Novex)/1:10 β-mercaptoethanol (Sigma) for 10 min. Samples were electrophoresed at 200 V constant on NuPAGE 4–12% Bis Tris Midi gels (Invitrogen) with NuPage MES-SDS running buffer (Invitrogen). The total volume of each sample containing 20 μg total protein was loaded on each lane. After electrophoresis, gels were incubated in 20% ethanol (Decon Laboratories) for 5 min at RT and electroblotted onto iBlot 2 NC Regular Stacks (Invitrogen) using the iBlot Dry Blotting preset 7 min blotting program for DSG cross-linking and/or the 20 V constant preset blotting program for GA cross-linking. The membrane was briefly rinsed in ultrapure water and incubated in 4% paraformaldehyde/PBS (Alfa Aesar) for 30 min at RT. Membranes were blocked in casein buffer 0.5% (BioRad) for 1 h at RT. After blocking, membranes were incubated with primary antibodies overnight at 4 °C. Membranes were briefly rinsed in PBS-Tween 0.1%, washed 3 × 10 min in PBS-Tween 0.1%, and incubated with the corresponding secondary LI-COR antibodies (1:20,000 in casein buffer/PBS 1:1/Tween 0.1%) at RT for 1 h in the dark. Membranes were rinsed in PBS-Tween 0.1%, washed 3 × 10 min in PBS-Tween 0.1%, and imaged on a LI-COR Odyssey CLx imaging system (preset Western analysis, auto scan, resolution 169 microns, scan speed fast, focus offset 0.0 mm, intensity auto). G51D samples were not analyzed blinded. Data signals of each band for all samples are provided in the source data file for Fig. 2A,B.

## Immunoprecipitation and mass spectrometry

Two different blood samples cross-linked with DSG (1.34 mM) were incubated and processed as described above. Samples were immunoprecipitated using the anti α-synuclein 211 antibody, mock samples were incubated with Protein G Agarose beads alone. We used 2 different α-synuclein samples in 1 technical replicate. Each of the α-synuclein samples was run on a SDS-PAGE and gel slices at the positions of the respective control Western blot were analyzed. Samples were isolated from human blood (samples n = 2, gel slice excised at 60 kDa), and controls were treated the same, but the immunoprecipitation was conducted without addition of antibody. After SDS-Page, gel pieces were lypholyzed and digested with trypsin prior to Mass spectrometry analysis. Mock analysis was used as a background substraction for the α-synuclein-211 samples.

## Size-exclusion chromatography

Samples were injected on a Superdex 200 Increase (10/300 GL) column (Cytivia) at room temperature and eluted with 50 mM NH4Ac (pH 7.4) while measuring (in-line) the conductivity and the 280-nm absorption of the eluate. For size estimation, a gel filtration standard (catalog no. 151-1901, Bio-Rad) was run on the column, and the calibration curve was obtained by semilogarithmic plotting of molecular weight versus the elution volume divided by the void volume.

## Mass spectrometry

Samples were analyzed on an ABI 4800 TOF/TOF Matrix-Assisted Laser Desorption Ionization (MALDI) mass spectrometer (Applied Biosystems, Foster City, CA). Samples undergoing trypsin digestion were incubated overnight in 50 mM $NH_4HCO_3$, 5 mM $CaCl_2$, and 12.5 ng/μL$^{-1}$ of trypsin, then desalted and concentrated using Millipore C18 ZipTips before spotting. Trypsin-digested samples and samples for intact mass analysis were prepared for spotting by mixing 0.5 μL of sample with 0.5 μL of α-cyano-4-hydroxy-trans-cinnamic acid (10 mgml$^{-1}$ in 70% acetonitrile, 0.1% TFA). Data were analyzed using the Mascot algorithm by searching against the updated nonredundant database from NCBI After drying, samples were rinsed with 0.1% TFA. ESI intact mass measurement was conducted on an Orbitrap LC-MS instrument. The cross-linked protein purified via immunoprecipitation and Size Exclusion Chromatography was desalted using ZipTip C18 columns to remove salts and other contaminants. The eluted protein was resuspended in 0.1% formic acid in water and loaded onto a C18 reverse-phase column for separation. Mass spectrometer calibration was performed using BSA. Databases used were: SwissProt_5712_rev(01/06/10) and SwissProt_2011_03_rev (03/23/11). All protein hits with at least 1 unique peptide (trypsin) hit were included.

## ELISA

For the ELISA analysis, supernatants and pellets from the blood samples were used. Samples were processed and analyzed as described before (Sanderson et al, 2020) using 2F12 as a capture, SOY1 as a sulfo-tagged detection antibody on an MSD ELISA platform and a Sector 2400 imager. Fractions measured were of total cytosolic α-synuclein (supernatant, soluble fraction) and membrane-associated α-synuclein (pellet after treatment with 1% Triton, Triton X-soluble fraction).

## Antibodies

Antibodies used were 2F12 (MABN1817, Merck, 1:2000 for Western blot analysis, 100 ng/20 μl proteinA/G beads for immunoprecipitation) and SOY1 (MABN1818, Merck) to α-synuclein, anti-DJ-1 (GeneTex, 1:2000), anti-hemoglobin (ab191183, Abcam, 1:1000), and anti-α-synuclein-211 (sc-12767, Santa Cruz). Secondary antibodies used were IRDye® 800CW Goat anti-Mouse IgG Secondary Antibody (against α-synuclein; P/N: 926-32210) and IRDye® 680RD Donkey anti-Rabbit IgG Secondary Antibody (against DJ1; P/N: 926-68073).

## Statistical analyses

Western Blot: the ratio of protein tetramers:monomers was analyzed using the Image Studio software western analysis according to the manufacturer's instructions (preset Western analysis, band markers were manually placed, background subtraction to reduce adjacent signal from smearing near the lanes was performed using median background substraction, border with 3, top/bottom). Data analysis was performed using GraphPad Prism 7 (GraphPad Software, La Jolla, CA, USA). Statistical significance was determined by Mann–Whitney-U test ($p < 0.05$, two-tailed). Samples are displayed as mean ± s.e.m. or s.d. Correlation analysis has been performed using Pearson's correlation coefficient. $P < 0.05$ was accepted as threshold for statistical

significance. Demographic and clinical measures were examined using two-tailed Welch's t-tests or Mann–Whitney-U tests for non-normally distributed data. $P < 0.05$, corrected for multiple comparisons was accepted as threshold for statistical significance. Analyses were performed in R (R-4.2.1; https://www.r-project.org/).

### The paper explained

#### Problem

α-synuclein is a naturally occurring protein in different tissues of our bodies. It exists in different forms: monomers, which are individual molecules, and multimers such as tetramers, which are small clusters of multiple α-synuclein molecules working together. These multimers are believed to have protective roles. In Parkinson's disease (PD), α-synuclein can undergo changes that lead to the formation of abnormal clumps called aggregates. Some aggregates are thought to be toxic and can harm brain cells, contributing to the development of PD symptoms. Understanding how α-synuclein transitions from its protective multimeric form to monomers and subsequently the harmful aggregates is a crucial area of research in PD. Unraveling this process might help to develop better ways to diagnose and treat this complex neurological disorder.

We used different techniques to analyze the relative levels and nature of the α-synuclein tetramer in blood from humans. Some of the humans have genetic mutations of the α-synuclein gene (SNCA): The G51D mutation in the SNCA gene is linked to PD. This mutation causes a change in the α-synuclein protein, making it more likely to misfold and form clumps in the brain. We also used blood from healthy humans as controls and blood from sporadic PD patients. Sporadic PD is a form of PD that occurs without a clear genetic or hereditary cause. In contrast to familial (genetic) PD, where there is a known genetic mutation that increases the risk of developing the disease and often runs in families, sporadic PD appears to occur without a strong genetic link.

#### Results

Combining an in vitro approach, our results show, that the relative levels and thus the ratio of α-synuclein tetramers in comparison to monomers are diminished in patients with familial (genetic) and sporadic PD. We could demonstrate, that the relative levels of α-synuclein tetramers to monomers are already reduced in carriers with G51D mutations of the SNCA gene not having developed clinical signs of PD. We were also interested whether the tetrameric form also includes any other proteins which would have been significant for the understanding of possible causes for patients with sporadic PD. We show that based on our results, the tetramer in blood only exists of α-synuclein and no other proteins.

#### Impact

Investigating α-synuclein monomers and tetramers in blood for sporadic PD could hold potential value for several reasons: Overall, the easy accessibility of blood makes it an important body fluid as a biomarker to monitor levels of disease for diagnostic or prognostic evaluation. Improved assays performed on larger cohorts for the detection of α-synuclein tetramers could determine whether α-synuclein tetramers serve as a diagnostic or prognostic biomarker with predictive capability. However, it's important to note that α-synuclein analysis in blood is a complex field, and several challenges need to be addressed. Additionally, variations between individuals and different techniques used for analysis can impact results. While investigating monomers and tetramers and other multimerss in blood for sPD is promising, further research and validation are necessary before these measures can be established as reliable biomarkers for routine clinical use. Nonetheless, ongoing studies in this area are essential for advancing our understanding of sPD and improving diagnostic and monitoring tools.

## For more information

ALZFORUM | NETWORKING FOR A CURE: https://www.alzforum.org/.

## Data availability

This study includes no data deposited in external repositories.

The source data of this paper are collected in the following database record: biostudies:S-SCDT-10_1038-S44321-024-00083-5.

## Peer review information

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

## Acknowledgements

We thank Eva Kess and Anja Schneider for their support and the helpful discussions. LdB was funded by the DZNE and University Bonn. ARR is funded by the "Maria Zambrano" fellowship and by Grant PID2021-126584OA-I00 funded by MCIN/AEI/10.13039/501100011033 and by "ERDF A way of making Europe". NH is funded by a grant by the Rosetrees and Stoneygate Trusts. UW is funded by the DFG, BMBF, dPV, DZNE and the University Bonn. LAL and RSW are supported by the Wellcome Trust: 205167/Z/16/Z, and from the National Institute of Health Research Biomedical Research Centre. TB was supported by grants from the UK Dementia Research Institute (DRI), which receives its funding from DRI Ltd., the UK Medical Research Council and Alzheimer's Society, the US National Institute of Neurological Disorders and Stroke grants (U54-NS110435, R01-NS109209, and R01-NS078165, the Eisai Pharmaceutical postdoctoral programme and the Chan Zuckerberg Collaborative Pairs Initiative Phase 2.

## Author contributions

**Laura de Boni**: Conceptualization; Data curation; Formal analysis; Validation; Investigation; Visualization; Methodology; Writing—original draft; Project administration; Writing—review and editing. **Amber Wallis**: Resources; Data curation; Formal analysis; Project administration; Writing—review and editing. **Aurelia Hays Watson**: Data curation; Formal analysis; Writing—review and editing. **Alejandro Ruiz-Riquelme**: Data curation; Formal analysis; Writing—review and editing. **Louise-Ann Leyland**: Resources; Data curation; Writing—review and editing. **Thomas Bourinaris**: Resources; Data curation; Writing—review and editing. **Naomi Hannaway**: Resources; Project administration; Writing—review and editing. **Ullrich Wuellner**: Resources; Supervision; Funding acquisition; Project administration; Writing—review and editing. **Oliver Peters**: Resources; Writing—review and editing. **Josef Priller**: Resources; Writing—review and editing. **Björn H Falkenburger**: Resources; Writing—review and editing. **Jens Wiltfang**: Resources; Writing—review and editing. **Mathias Bähr**: Resources; Writing—review and editing. **Inga Zerr**: Resources; Writing—review and editing. **Katharina Buerger**: Resources; Writing—review and editing. **Robert Perneczky**: Resources; Writing—review and editing. **Stefan Teipel**: Resources; Writing—review and editing. **Matthias Löhle**: Resources; Writing—review and editing. **Wiebke Hermann**: Resources; Writing—review and editing. **Björn-Hendrik Schott**: Resources; Writing—review and editing. **Kathrin Brockmann**: Resources; Writing—review and editing. **Annika Spottke**: Resources; Writing—review and editing. **Katrin Haustein**: Data curation; Formal analysis; Writing—review and editing. **Peter Breuer**: Resources; Data curation; Writing—review and editing. **Henry Houlden**: Resources; Data curation; Writing—review and editing. **Rimona S Weil**: Resources; Data curation; Supervision; Funding acquisition; Project administration; Writing—review and editing. **Tim Bartels**: Conceptualization; Resources; Data curation; Formal analysis; Supervision; Funding acquisition; Validation; Investigation; Visualization; Methodology; Project administration; Writing—review and editing.

Source data underlying figure panels in this paper may have individual authorship assigned. Where available, figure panel/source data authorship is listed in the following database record: biostudies:S-SCDT-10_1038-S44321-024-00083-5.

## Disclosure and competing interests statement

UW received consulting fees/contracting fees from STADA Pharm, Idorsia, Zambon, Abbvie, Bayer, Bial, and Roche. RSW has received speaking honoraria from GE Healthcare and writing honoraria from Britannia. RSW has received fees for consulting for Therakind. TB received consulting/contracting fees from Jennsen and Arvinas, part of the research was conducted by research funds provided Eisai Pharmaceuticals.

# Expanded View Figures

**A**

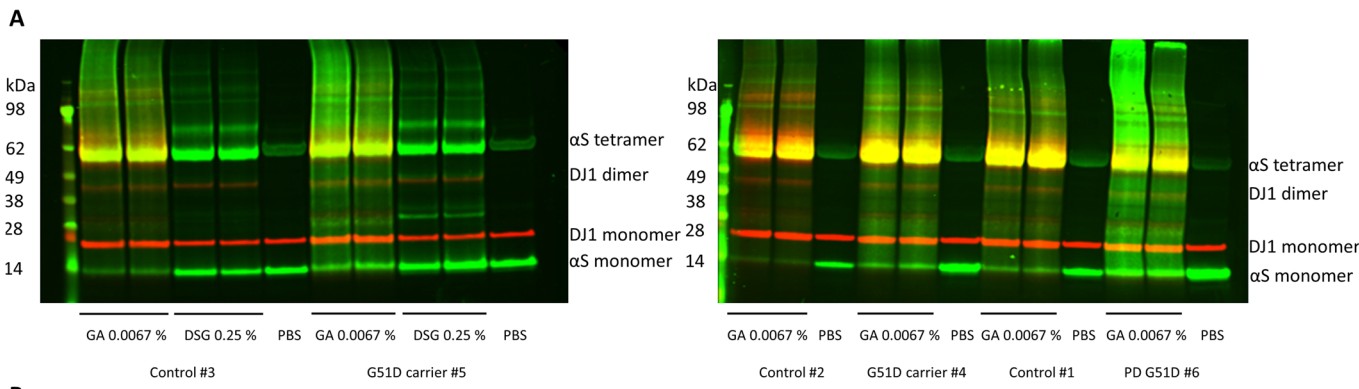

**B**

## LC-MS, 60 kDa protein

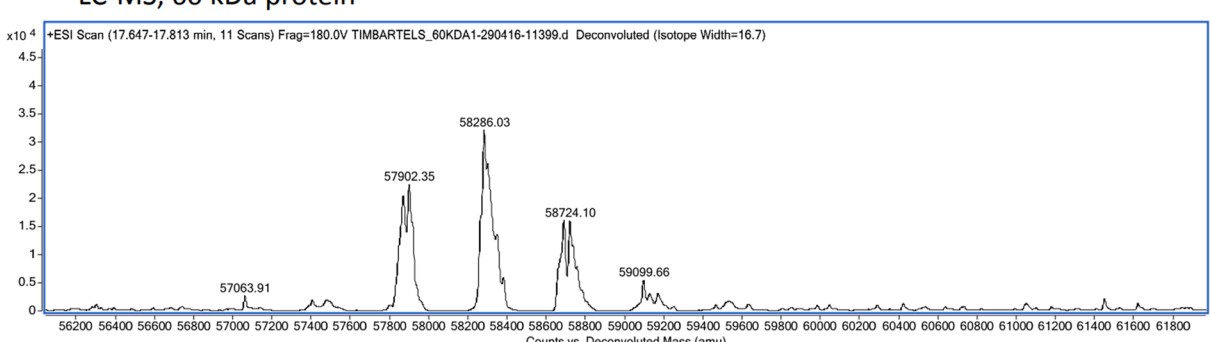

**Figure EV1.  Representative Western blot and Mass spectrometry analysis.**

(**A**) Western blot analysis of G51D carriers and one PD G51D patient blood lysate compared to controls. Representative pictures of Western blot analyses from controls, G51D carriers and one PD patient with a G51D mutation using the cross-linker GA and DSG. All samples have been analyzed in technical duplicates. Information on clinical characteristics is provided in Table 1. Samples for Fig. EV1 (depleting hemoglobin, cross-linking, gels, blots) were processed in parallel on different blots due to the samples size. No loading controls were run on the western blot as the full volume of each processed sample containing 20 μg of total protein was loaded into each gel pocket. (**B**) Mass spectrometry traces of immunoprecipitated blood-derived α-synuclein. The graph displays the expected mass for isolated tetrameric α-synuclein: 58,286 kDa. In comparison, the expected mass for the DSG cross-linker (unconjugated) would be 326 kDa. Source data are available online for this figure.

## Loss of Western blot signal after different time points of freezing

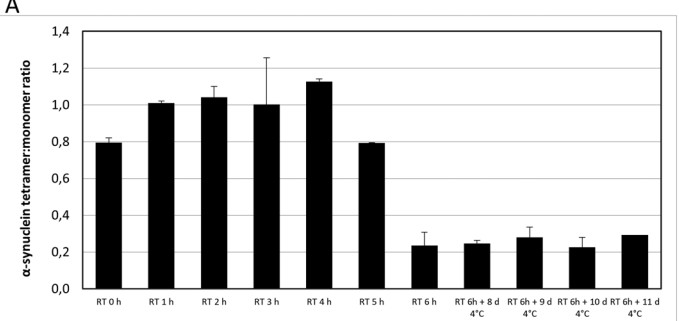

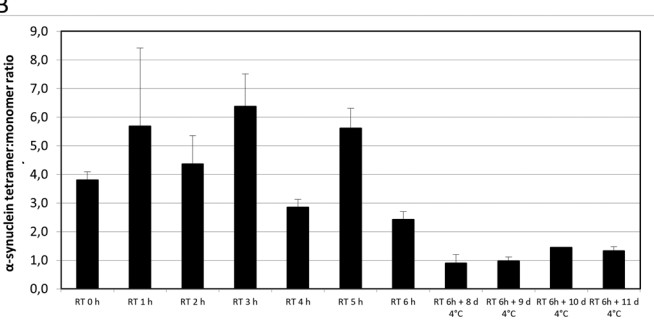

## Signal intensities remain stable after multiple freeze thaw cycles

## Hemoglobin intensities are not correlated with tetramer:monomer ratios

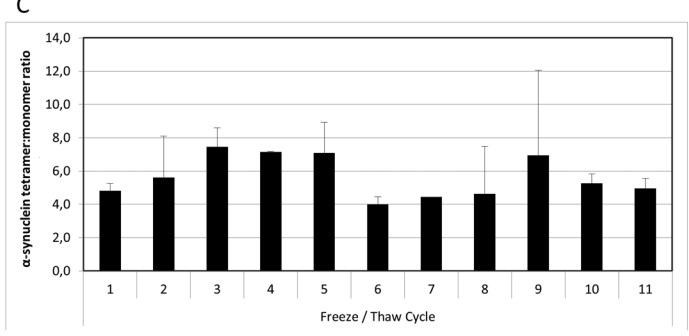

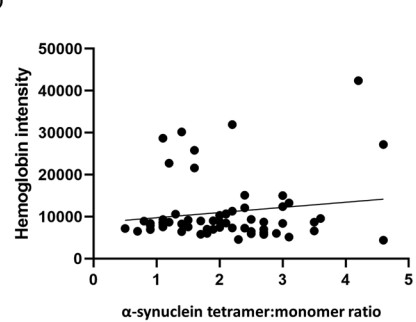

**Figure EV2.  Validation of the cross-linking protocol.**

The procedure is described in the Methods section. (**A**) Quantification of the Western blot of cross-linked (DSG) human whole blood after Hemoglobin depletion (ratio Blood:HemogloBind 1:1, $n = 1$, control sample). Blood samples were left at room temperature for different time points (0–6 h) or placed at 4 °C for 8–11 days after the samples have been kept at 6 h RT. Samples were analyzed in two technical replicates. Due to degradation processes, the signal/ratio drops after 6 h at RT and subsequent cooling at 4 °C. Data is displayed as mean ± s.d. (**B**) Quantification of the Western blot of cross-linked (DSG) human whole blood after Hemoglobin depletion (ratio Blood:HemogloBind 1:4, $n = 1$, control sample). Blood samples were left at room temperature for different time points (0–6 h) or placed at 4 °C for 8–11 days after the samples have been kept at 6 h RT. Samples were analyzed in two technical replicates. Due to degradation processes, the signal/ratio drops after 6 h at RT and subsequent cooling at 4 °C. Data is displayed as mean ± s.d. (**C**) Signal intensities of the Western blot analysis remain stable after multiple freeze/thaw cycles (Blood:HemogloBind 1:1, $n = 1$, control sample). Samples were analyzed in two technical replicates. Data is displayed as mean ± s.d. (**D**) Hemoglobin intensities after removal of Hemoglobin were correlated (Pearson correlation) with α-synuclein tetramer:monomer ratios (Blood:HemogloBind 1:1, $r = 0.2$, $p = 0.2$, $n = 60$). RT room temperature, DSG Disuccinimidyl glutarate.

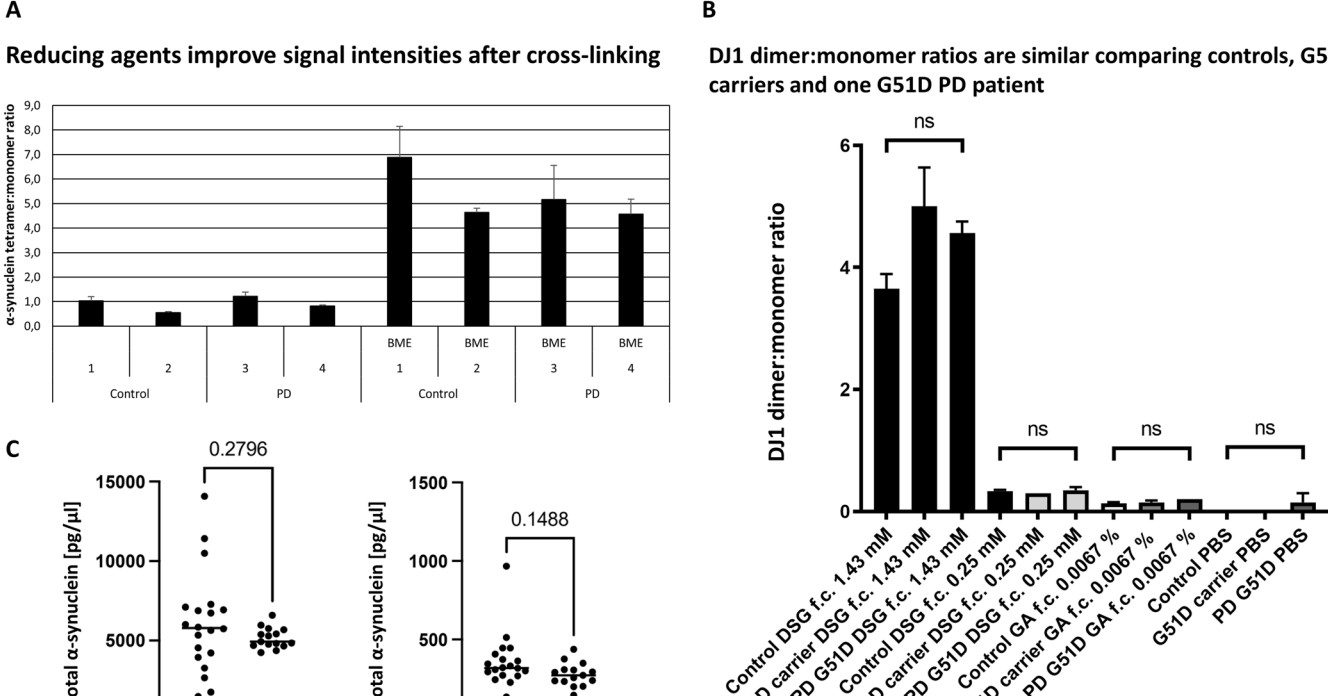

**A**

**Reducing agents improve signal intensities after cross-linking**

**B**

**DJ1 dimer:monomer ratios are similar comparing controls, G51D carriers and one G51D PD patient**

**C**

**Figure EV3. Validation of the cross-linking protocol.**

(A) Validation of the cross-linking protocol. The procedure is described in the Methods section. Quantification of the Western blot of cross-linked (DSG) human whole blood after Hemoglobin depletion (ratio Blood:HemogloBind 1:1, $n = 4$, 2 control samples, 2 PD samples). The signal is increased after reducing the sample at 70 °C using 4 × NuPage LDS sample buffer (Novex)/1:10 β-mercaptoethanol (Sigma). Samples were analyzed in two technical replicates. Data is displayed as mean ± s.d. BME = β-mercaptoethanol, PD = Parkinson's disease, DSG=Disuccinimidyl glutarate. (B) DJ1 dimer:monomer ratios are similar between controls, G51D carriers and PD G51D patients. The DJ1 protein serves as an internal control for the cross-linking procedure. All groups were compared using Mann–Whitney-U test. G51D carriers $n = 2$, G51D PD patient ($n = 1$), controls $n = 3$. Samples were analyzed in two technical replicates. Data is displayed as mean ± s.d. F.c. = final concentration, PD = Parkinson's disease, DSG = Disuccinimidyl glutarate, GA = glutaraldehyde. (C) Analysis of total α-synuclein blood levels. Left, total, blood-derived cytosolic α-synuclein levels do not differ between sPD patients ($n = 20$) and controls ($n = 15$, $p = 0.3$). Cytosolic, soluble α-synuclein was derived after mechanical cell lysis. Right, total, blood-derived membrane-associated α-synuclein levels do not differ between sPD patients ($n = 20$) and controls ($n = 15$, $p = 0.2$). Membrane-associated α-synuclein was derived after mechanical cell lysis from the 1% Triton-soluble fraction. All Samples were analyzed in two technical replicates. All groups were compared using Mann–Whitney-U test. Mean for each sample is displayed.

## Significant correlation between Summary Cognitive Score but not gender, UPDRS motor score, MoCA, MMSE, or hallucinator scale and α-synuclein tetramer:monomer ratios

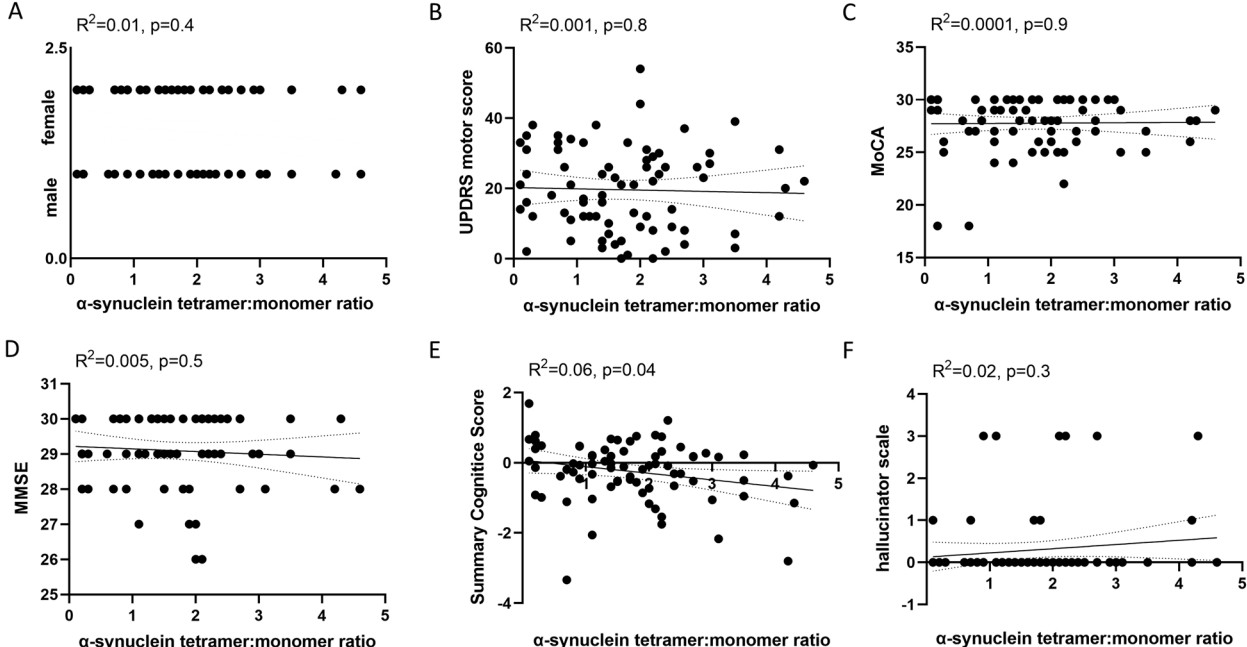

**Figure EV4.  Correlation analysis of α-synuclein tetramer:monomer ratios and clinical parameters.**

Pearson correlation of cohort 1 shows Pearson's correlation analysis of α-synuclein tetramer:monomer ratios and (**A**) gender, (**B**) UPDRS motor score, (**C**) MoCA, (**D**) MMSE, (**E**) Summary Cognitive Score, (**F**) hallucinator scale.

**No significant correlation between gender, UPDRS motor score, MoCA or MMSE and α-synuclein tetramer:monomer ratios**

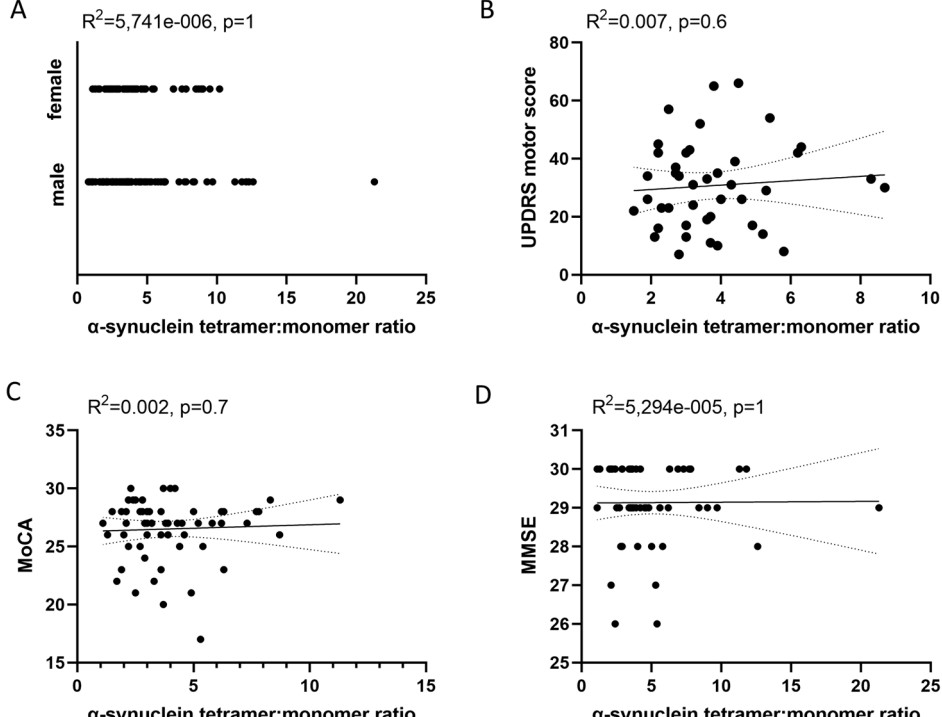

**Figure EV5. Correlation analysis of α-synuclein tetramer:monomer ratios and clinical parameters.**

Pearson correlation of cohort 2 shows no significant correlation of α-synuclein tetramer:monomer ratios and (A) gender, (B) UPDRS motor score, (C) MoCA, (D) MMSE.

