## [Peer Review File · EMBO Molecular Medicine]

Aggregation-resistant alpha-synuclein tetramers are reduced in the blood of Parkinson's patients

Laura de Boni, Amber Wallis, Aurelia Hays-Watson, Alejandro Ruiz-Riquelme, Louise-Ann Leiland, Thomas Bourinaris, Naomi Hannaway, Ullrich Wüllner, Oliver Peters, Josef Priller, Bjoern Falkenburger, Jens Wiltfang, Mathias Bähr, Inga Zeer, Katharina Buerger, Robert Perneczky, Stefan Teipel, Matthias Loehle, Wiebke Hermann, Bjoern Schott, Kathrin Brockmann, Annika Spottke, Katrin Hausteil, Peter Breuer, Henry Houlden, Rimona Weil, and Tim Bartels

Corresponding author: Tim Bartels (t.bartels@ucl.ac.uk)

Review Timeline:

Submission Date:	5th Feb 24
Editorial Decision:	14th Mar 24
Revision Received:	4th Apr 24
Editorial Decision:	25th Apr 24
Revision Received:	2nd May 24
Accepted:	17th May 24

Editor: Poonam Bheda

Transaction Report:

14th Mar 2024

Dear Dr. Bartels,

Thank you again for submitting your work to EMBO Molecular Medicine. We have now heard back from the two reviewers who agreed to evaluate your study. As you will see below, the reviewers appreciate that the presented approach addresses a relevant problem. However, they raise some concerns, which we would ask you to address in a major revision.

I think that the recommendations of the reviewers are rather clear and I therefore do not see the need to repeat the comments listed below. A fundamental point raised refers to the need for further support to ensure the accuracy of the alpha-synuclein tetramer to monomer ratio, given the smears observed in the cross-linked samples as pointed out by Reviewer 1.

Addressing the reviewers' concerns in full in a point-by-point response will be necessary for further considering the manuscript in our journal, and acceptance of the manuscript will entail a second round of review. EMBO Molecular Medicine encourages a single round of revision only and therefore, acceptance or rejection of the manuscript will depend on the completeness of your responses included in the next, final version of the manuscript. For this reason, and to save you from any frustrations in the end, I would strongly advise against returning an incomplete revision. If you would like to discuss further the points raised by the referees, I am available to do so via email or video. Let me know if you are interested in this option.

We are expecting your revised manuscript within three months, if you anticipate any delay, please contact us. When submitting your revised manuscript, please carefully review the instructions that follow below. We perform an initial quality control of all revised manuscripts before re-review; failure to include requested items will delay the evaluation of your revision.

We require:

4) A .docx formatted letter INCLUDING the reviewers' reports and your detailed point-by-point responses to their comments. As part of the EMBO Press transparent editorial process, the point-by-point response is part of the Review Process File (RPF), which will be published alongside your paper.

5) A complete author checklist, which you can download from our author guidelines (<https://www.embopress.org/page/journal/17574684/authorguide#submissionofrevisions>). Please insert information in the checklist that is also reflected in the manuscript. The completed author checklist will also be part of the RPF.

6) Please note that all corresponding authors are required to supply an ORCID ID for their name upon submission of a revised manuscript.

7) It is mandatory to include a 'Data Availability' section after the Materials and Methods. Before submitting your revision, primary datasets produced in this study need to be deposited in an appropriate public database, and the accession numbers and database listed under 'Data Availability'. Please remember to provide a reviewer password if the datasets are not yet public (see <https://www.embopress.org/page/journal/17574684/authorguide#dataavailability>).

This study includes no data deposited in external repositories.

8) For data quantification: please specify the name of the statistical test used to generate error bars and P values, the number (n) of independent experiments (specify technical or biological replicates) underlying each data point and the test used to calculate p-values in each figure legend. The figure legends should contain a basic description of n, P and the test applied. Graphs must include a description of the bars and the error bars (s.d., s.e.m.). Please provide exact p values.

9) Our journal encourages inclusion of *data citations in the reference list* to directly cite datasets that were re-used and obtained from public databases. Data citations in the article text are distinct from normal bibliographical citations and should

directly link to the database records from which the data can be accessed. In the main text, data citations are formatted as follows: "Data ref: Smith et al, 2001" or "Data ref: NCBI Sequence Read Archive PRJNA342805, 2017". In the Reference list, data citations must be labeled with "[DATASET]". A data reference must provide the database name, accession number/identifiers and a resolvable link to the landing page from which the data can be accessed at the end of the reference. Further instructions are available at .

13) Author contributions: CRedit has replaced the traditional author contributions section because it offers a systematic machine readable author contributions format that allows for more effective research assessment. Please remove the Authors Contributions from the manuscript and use the free text boxes beneath each contributing author's name in our system to add specific details on the author's contribution. More information is available in our guide to authors.

Please also suggest a striking image or visual abstract to illustrate your article as a PNG file 550 px wide x 300-600 px high. Share synopsis text and image, as well as eTOC:

Please note that these would be the final versions and changes during proofing are usually not allowed

16) As part of the EMBO Publications transparent editorial process initiative (see our policy here: https://www.embopress.org/transparent-process#Review_Process), EMBO Molecular Medicine will publish online a Peer Review File (PRF) to accompany accepted manuscripts.

In the event of acceptance, this file will be published in conjunction with your paper and will include the anonymous referee reports, your point-by-point response and all pertinent correspondence relating to the manuscript. Let us know whether you agree with the publication of the PRF and as here, if you want to remove or not any figures from it prior to publication.

I look forward to receiving your revised manuscript.

Yours sincerely,

Poonam Bheda

Poonam Bheda, PhD
Scientific Editor
EMBO Molecular Medicine

**** Reviewer's comments ****

Referee #1 (Remarks for Author):

In this manuscript, de Boni et al. presented their study testing whether or not the ratio of tetrameric and monomeric forms of alpha-synuclein in the blood could be used for clinical diagnostic purposes. The method involves chemical cross-linking followed by Western blotting against anti-alpha-synuclein antibodies. The premise is tetrameric alpha-synuclein is aggregation-resistant, and monomeric alpha-synuclein is aggregation-prone and might be pathogenic. The data supported that Parkinson's disease patients showed a lower tetramer: monomer ratio than healthy controls. However, no significant differences were observed in disease duration or age. While this idea is interesting, and its potential for developing a novel diagnostic should be examined, the data shows its limitations. In addition to the potential problems that the authors listed, my main concern with using this method as a potential diagnostic tool lies in Figures 2A & B, in which the lanes of the cross-linked sample consisted of significant smearing contiguous with the band corresponding to the tetrameric alpha-synuclein. The smears are likely higher oligomeric species of alpha-synuclein, which may be drawn from the tetrameric or monomeric pool and thus would likely skew the ratio in unpredictable ways. They may also be alpha-synuclein having cross-linked with other proteins in the lysate and thus introducing more uncertainty. Moreover, the presence of the smear and the inherent heterogeneity of the band patterns could introduce significant errors in the ratio. For example, the smear adjacent to the tetrameric band has about the same intensity as the band of monomeric aSyn in lane 1.34 of Figure 2A and so the ratio could be significantly artificially elevated, especially in the control lanes where the monomeric bands are relatively less intense.

Below are comments that I hope will help the authors improve the manuscript.

Line 87 - 88. "... plays a crucial role in the regulation of synaptic function...", please specify the functions and cite the primary sources. If no specific functions have been identified, then perhaps reword to something of the effect "... it is believed to play a crucial role ...".

Line 87. "... primarily in neurons" then in line 95, "...primarily in red blood cells..". Perhaps reword to avoid confusion.

Line 93. "...physiological and pathological a-synuclein...". Perhaps define physiological and pathological forms of a-synuclein and reference primary data to avoid confusion of physiological with biologically functional form and pathological with disease-causing form. If still unclear about the forms of aSyn in these states, perhaps rephrase to reflect this.

Line 99. Define "...native form"

Line 99. Consider replacing "unfolded" with 'disordered' or 'random coil' or 'unstructured' as 'unfolded' can be confused with the unfolding of a folded protein.

Line 100 - 101. "...equilibrium with a helically-folded ...". Consider rephrasing unless there is clear data showing that an equilibrium exists between these two forms. Or define what you meant by 'equilibrium'.

Line 122. "...misfolded a-synuclein..". Define the term "misfolded" used here.

Line 147. Again, define "physiological". Is it the same as a native or functional form?

Line 160. Perhaps define "best characterized" or rephrase since the fibril structures are currently the best characterized.

Line 209 - 210. Perhaps replace "equilibrium" with "ratio"

Line 221. See above (line 209)

Line 432. "... lysate", please examine; I didn't see a lysate produced in the procedure described.

Line 458. Please specify the "LI-COR antibodies" and include a description and perhaps the product number.

Referee #2 (Comments on Novelty/Model System for Author):

I believe that this study is of great importance and translatability. First, the physiological role of alpha-synuclein tetramers is not yet fully understood. Second, we really need usable biomarkers of alpha-synuclein pathology in the blood, both for diagnostic purposes and to monitor the efficacy of therapies that counteract the disease.

Referee #2 (Remarks for Author):

I believe that this study is of great importance and translatability. First, the physiological role of alpha-synuclein tetramers is not yet fully understood. Second, we really need usable biomarkers of alpha-synuclein pathology in the blood, both for diagnostic purposes and to monitor the efficacy of therapies that counteract the disease. I do propose just minor revisions:

as a general suggestion for introduction it can be better and more clearly stated that total soluble alpha-synuclein reflects three main states: it can be moderately decreased due to synucleinopathy, increased due to synaptic damage, and increased due to blood contamination. Therefore, it is very ineffective as a biomarker of a single disease process. Seed amplification assays have been successfully applied to detect misfolded alpha-synuclein, but there is only one study conducted on blood, which many groups are trying to replicate without success. In addition, SAAs are very delicate techniques, very unstable, and few groups perform them successfully on a routine basis.

line 144 Serum albumin is an interactor of alpha-synuclein (PMID: 31034772), but lipoproteins might interact more with the aggregation kinetics of SAAs, making them inapplicable in clean plasma/serum (PMID: 37005644). The alpha-synuclein/lipoprotein interaction is already relevant in CSF and even more so in blood, where lipoproteins are about 30 times more concentrated. In fact, the only published positive result on SAA in serum was obtained by immunoprecipitating alpha-synuclein before applying SAA.

I hope that this line of research will be continued and that the ratio of tetrameric to monomeric alpha-synuclein will be tested on larger cohorts. It will be necessary, however, for the experimental procedures for measurement to be further standardized so that they can be easily replicated by other groups.

Referee #1 (Remarks for Author):

In this manuscript, de Boni et al. presented their study testing whether or not the ratio of tetrameric and monomeric forms of alpha-synuclein in the blood could be used for clinical diagnostic purposes. The method involves chemical cross-linking followed by Western blotting against anti-alpha-synuclein antibodies. The premise is tetrameric alpha-synuclein is aggregation-resistant, and monomeric alpha-synuclein is aggregation-prone and might be pathogenic. The data supported that Parkinson's disease patients showed a lower tetramer: monomer ratio than healthy controls. However, no significant differences were observed in disease duration or age. While this idea is interesting, and its potential for developing a novel diagnostic should be examined, the data shows its imitations.

The thank the reviewer for her/his evaluation of our work and valuable feedback.

In addition to the potential problems that the authors listed, my main concern with using this method as a potential diagnostic tool lies in Figures 2A & B, in which the lanes of the cross-linked sample consisted of significant smearing contiguous with the band corresponding to the tetrameric alpha-synuclein. The smears are likely higher oligomeric species of alpha-synuclein, which may be drawn from the tetrameric or monomeric pool and thus would likely skew the ratio in unpredictable ways. They may also be alpha-synuclein having cross-linked with other proteins in the lysate and thus introducing more uncertainty. Moreover, the presence of the smear and the inherent heterogeneity of the band patterns could introduce significant errors in the ratio. For example, the smear adjacent to the tetrameric band has about the same intensity as the band of monomeric aSyn in lane 1.34 of Figure 2A and so the ratio could be significantly artificially elevated, especially in the control lanes where the monomeric bands are relatively less intense.

Yes, crosslinking can lead to smearing on Western blot membranes. We agree that this can introduce bias. According to Dettmer et al. (Dettmer *et al*, 2013), the ideal concentration for DSG crosslinking falls between 2-3 mM. Utilizing DSG at 5 mM, especially with intact cells, significantly reduces or entirely eliminates α -synuclein bands. For *in vitro* crosslinking, we found that signal strength decreases when DSG exceeds 2.9 mM, while crosslinking efficiency drops at concentrations below 0.7 mM ((Boni *et al*, 2022); see Fig. 1).

Fig. 1

Fig. 1 demonstrates the gradual changes in western blot signals using different amounts of the crosslinker DSG for *in vitro* crosslinking in brain lysates (Boni *et al*, 2022).

To prevent overcrosslinking or undercrosslinking (meaning that we keep a ratio where both monomer and tetramer are in the dynamic range of the intensity quantification), a concentration of 1.34 mM DSG was selected despite some smearing in the lanes. Nonetheless, using DSG at concentrations of 0.25 mM or 1.34 mM in blood samples, controls show higher tetramer to monomer ratios compared to G51D patients, as seen in Fig. 2 of the main manuscript.

With regard to smearing, which is more prominent when using higher crosslinking concentrations, we applied a stringent analysis protocol, subtracting a higher threshold background and thereby the smear. Details on the analysis are now extended in the material and methods section (line 516-517, line 565-568) and mentioned smearing in the limitations section (line 377). In detail here:

We configured the LICOR imaging system with the following settings: the analysis type was set to Western analysis, with the membrane preset and auto scan selected for all membranes. The auto scan feature was chosen because it offers a broader dynamic range compared to other scanning options. We processed a maximum of four membranes simultaneously. The resolution was consistently set to 169 microns, as this is recommended for Western blots. The scan speed was adjusted to the fastest setting. The focus was fixed at 0.0 mm, and the intensity was configured to auto.

For band analysis using Image Studio, we selected the Western preset, then manually adjusted the boundaries around all lanes on a single membrane and specified the number of lanes. We switched to a single color (one channel only) mode and manually placed band markers. A crucial step was performing background subtraction with the median top and bottom background subtraction method. Figure 2 illustrates the band markers with yellow squares, including smaller squares at the top and bottom of each band marker designated for background subtraction. The width of the top and bottom squares was set to 3 (Fig. 3).

Fig. 2

Image from <https://www.licor.com/bio/support/contents/software/image-studio/analysis/gel-and-blot-background.html#image-studio-background-blots-gels>

Fig. 3

Fig. 3 shows Median top/bottom subtraction set to width 3 in a blot used for Fig. 2 of the main manuscript.

Direct background subtraction above and below the band of interest is crucial for eliminating adjacent smearing. We refrained from using a generalized background subtraction approach that targets arbitrary areas on the membrane. Opting out of background subtraction entirely results in the software assigning NA (not available) values for intensity measurements. We reevaluated our samples by employing smaller (top and bottom 1) and larger (top and bottom 5, Fig. 4) areas for background subtraction.

Fig. 4

Fig. 4 demonstrates the analysis using median background subtraction top/bottom 5. Even with larger areas for background subtraction, no adjacent bands are included in the analysis.

The intensity values fluctuated between -57 and 188 when comparing analyses using top and bottom 3 to those using top and bottom 1 (Table 1). Additionally, intensity values ranged from -108 to 37 when contrasting the use of top and bottom 3 with top and bottom 5 (Table 1). Overall, enlarging the area for background subtraction tends to slightly lower the number for intensity values of the bands. Nonetheless, this adjustment does not significantly alter the tetramer to monomer ratios. For 20 out of 32 ratios, there were no intensity differences observed in the ratios when comparing both the smaller and larger settings (top and bottom 1 or 5) to the initially used settings (top and bottom 3).

Table 1 Comparisons between different background substractions

Image Name	Channel	Lane Name	Cross-linker [concentration]	Band kDa	Difference signal intensity top/bottom 3 minus top/bottom 1	Difference signal intensity top/bottom 3 minus top/bottom 5	Difference tetramer:monomer ratio top/bottom 3 minus top/bottom 1	Difference tetramer:monomer ratio top/bottom 3 minus top/bottom 5
0000747_01	800	Control #2	DSG 0.25mM	60	17	0	0.0	0.0
0000747_01	800			14	-4	0		
0000747_01	800			60	68	-43	0.0	-0.1
0000747_01	800			14	9	9		
0000747_01	800		DSG 1.34mM	60	13	-54	1.1	-0.2
0000747_01	800			14	-4	0		
0000747_01	800			60	39	-65	-1.0	1.9
0000747_01	800			14	7	-15		
0000747_01	800	G51D carrier #4	DSG 0.25mM	60	-20	-9	0.0	0.0
0000747_01	800			14	4	-4		
0000747_01	800			60	-33	5	0.0	0.0
0000747_01	800			14	38	-55		
0000747_01	800		DSG 1.34mM	60	25	-33	-0.4	0.1
0000747_01	800			14	47	-12		
0000747_01	800			60	83	-108	0.0	0.3
0000747_01	800			14	4	-22		
0000747_01	800	Control #1	DSG 0.25mM	60	36	-31	0.0	0.0
0000747_01	800			14	0	0		
0000747_01	800			60	26	-12	0.2	0.0

0000747_01	800		DSG 1.34mM	14	-9	0	-0.9	-1.4
0000747_01	800			60	68	-25		
0000747_01	800			14	4	4		
0000747_01	800			60	7	-12	0.0	1.0
0000747_01	800			14	0	-9		
0000747_01	800	PD G51D #6	DSG 0.25mM	60	-42	12	0.0	0.0
0000747_01	800			14	16	37		
0000747_01	800			60	6	11	0.0	0.0
0000747_01	800			14	-42	36		
0000747_01	800		DSG 1.34mM	60	188	-83	0.0	0.0
0000747_01	800			14	34	-15		
0000747_01	800			60	-57	-102	-0.1	0.0
0000747_01	800			14	13	-32		

Furthermore, we adjusted the brightness and contrast for the visual presentation of all bands in a representative figure within the main manuscript (main manuscript Fig. 2). It's important to note that for analytical purposes, we did not enhance the brightness and contrast as we did for figure creation; in some cases, we employed grayscale to improve band visibility for analysis. Adjusting brightness or contrast does not affect the quantification values; such adjustments merely alter the representation of raw image pixels for display purposes. The quantified signal is derived from the total sum of individual pixel intensities within a defined band or lane box, minus the product of the background intensity value and its area. Therefore, while the overall appearance of an image can be modified through brightness or contrast adjustments, the underlying pixel intensities, and consequently the quantification results, remain unchanged.

Furthermore, we wanted to demonstrate the feasibility of different crosslinkers, unless giving different absolute intensity values. However, DSG did not give significant results in the G51D cohort. With regard to the smearing bias, we therefore re-analyzed a few samples from the German cohort using the crosslinker DSG (23 controls and 21 sPD) and the aforementioned settings for background subtraction. We can show, that upon usage of DSG, the tetramer:monomer ratio is still significantly lowered, reproducing the results obtained with GA (Fig. 4).

Fig. 4

Fig. 4 Cohort 2 re-analyzed with DSG instead of GA

We did not analyze any other multimers such as 80 or 100 kDa bands (Dettmer *et al*, 2013), as they were not present in all samples. Band markers were set to analyze tetrameric bands only (Fig. 5). We mentioned this bias in the discussion section (line 224, line 262-263, line 382-384, line 386).

Fig. 5

Fig. 5 shows the analysis on G51D and control samples using the median top/bottom 3 background subtraction. Other multimeric bands were not included in the analysis and background subtraction was placed above or below any other bands.

The other point raised in the reviewer's comment is about possible-protein interactions of the multimeric α -synuclein forms. Different experiments have been performed to detect possible protein-protein interactions (e.g. two-dimensional gel analyses on cross-linked α -synuclein species, analyzing differentially tagged α -synuclein molecules, mass spectrometry, immunoprecipitation followed by Western blotting, ion mobility-mass spectrometry and native top-down electron capture dissociation fragmentation). The data so far reveals that the tetrameric or dimeric α -synuclein represents primarily a homotetramer (Fernández und Lucas 2018; Wang et al. 2011; Dettmer et al. 2013; Jeacock et al. 2023; Dettmer et al. 2013). So far, we did not detect any interactions of the α -synuclein multimers with other proteins previously described by other laboratories to interact such as tau, anti- β tubulin III, anti-heat shock protein 70, anti-14-3-3 beta, and anti-HSP90 (Boni et al. 2022). For this study, we performed mass spectrometry analysis which is summarized in Table EV1 and line 225-228, line 231-233. Summary of mass spectrometry data of blood-derived 60 kDa α -synuclein after background subtraction (α -synuclein and mock samples) confirmed the data, that the α -synuclein tetramer in blood is likely a homomultimer. However, we still might miss transient protein interactions. We referred to this issue in the discussion section (line 363-365, line 386-389).

Below are comments that I hope will help the authors improve the manuscript.

The entire introduction has been revised according to the suggestions below (line 89-132).

Line 87 - 88. "... plays a crucial role in the regulation of synaptic function...", please specify the functions and cite the primary sources. If no specific functions have been identified, then perhaps reword to something of the effect "... it is believed to play a crucial role ...".

The functions have been specified and primary sources added (line 89-123).

Line 87. "... primarily in neurons" then in line 95, "...primarily in red blood cells..". Perhaps reword to avoid confusion.

The word "primarily" has been omitted (line 130-132).

Line 93. "...physiological and pathological a-synuclein...". Perhaps define physiological and

pathological forms of a-synuclein and reference primary data to avoid confusion of physiological with biologically functional form and pathological with disease-causing form. If still unclear about the forms of aSyn in these states, perhaps rephrase to reflect this.

We agree that the terms physiological, pathological, native, toxic, multimer, tetramer or oligomer can be confusing. In the context of α -synuclein, even an excess of physiological monomers due to SNCA duplication or triplications can be detrimental. We therefore agree, that we have to revise the introduction (line 89-128, line 166, line 179, line 196, line 207) as we clearly make a distinction between physiological multimers and pathological oligomers.

Line 99. Define "...native form"

The introduction has been revised. The term native is omitted. Instead, we focus on the conformational changes of the α -synuclein protein (line 89-128). See also abstract line 75, line 78)

Line 99. Consider replacing "unfolded" with 'disordered' or 'random coil' or 'unstructured' as 'unfolded' can be confused with the unfolding of a folded protein.

Unfolded has been replaced with disordered throughout the manuscript (e.g. line 203, line 211).

Line 100 - 101. "...equilibrium with a helically-folded ...". Consider rephrasing unless there is clear data showing that an equilibrium exists between these two forms. Or define what you meant by 'equilibrium'.

We rephrased the paragraph: Based on current understanding, the assembly of α -synuclein into multimers is critical for maintaining a ratio between aggregation-prone monomers and aggregation-resistant tetramers. A change of this ratio with bias towards monomeric forms is potentially linked to the initiation of disease (line 110-113).

Line 122. "...misfolded a-synuclein..". Define the term "misfolded" used here.

We explained that the disordered α -synuclein monomers can form β -sheet-like oligomers, fibrils and intracellular inclusions known as Lewy bodies, Lewy neurites or glial cytoplasmic inclusions. We omitted the term misfold as it is unprecise in this context (line 114-123, line, 166-167, line 179-180).

Line 147. Again, define "physiological". Is it the same as a native or functional form?

See revised introduction (line 124-128).

Line 160. Perhaps define "best characterized" or rephrase since the fibril structures are currently the best characterized.

See revised introduction. The term "best characterized" has been replaced by well characterized (line 210).

Line 209 - 210. Perhaps replace "equilibrium" with "ratio"

Equilibrium has been replaced with ratio (e.g. line 200, line 217, line 219, line 259, line 271).

Line 221. See above (line 209)

Equilibrium has been replaced with ratio (introduction and results, the latter line 217, 219, 259)

Line 432. "... lysate", please examine; I didn't see a lysate produced in the procedure described.

Blood suspensions were lysed by sonication. We added additional information to the material and methods section now described as crosslinking of lysate (line 486-488).

Line 458. Please specify the "LI-COR antibodies" and include a description and perhaps the product number.

The LI-COR antibodies refer to the secondary antibodies. Information has been added to the material and methods section. We used IRDye® 800CW Goat anti-Mouse IgG Secondary Antibody for α -synuclein, and IRDye® 680RD Donkey anti-Rabbit IgG Secondary Antibody for DJ1 (line 559-562)

References

- Boni, Laura de; Watson, Aurelia Hays; Zaccagnini, Ludovica; Wallis, Amber; Zhelcheska, Kristina; Kim, Nora et al. (2022): Brain region-specific susceptibility of Lewy body pathology in synucleinopathies is governed by α -synuclein conformations. In: *Acta neuropathologica* 143 (4), S. 453–469. DOI: 10.1007/s00401-022-02406-7.
- Dettmer, Ulf; Newman, Andrew J.; Luth, Eric S.; Bartels, Tim; Selkoe, Dennis (2013): In vivo cross-linking reveals principally oligomeric forms of α -synuclein and β -synuclein in neurons and non-neural cells. In: *The Journal of biological chemistry* 288 (9), S. 6371–6385. DOI: 10.1074/jbc.M112.403311.
- Fernández, Ricardo D.; Lucas, Heather R. (2018): Isolation of recombinant tetrameric N-acetylated α -synuclein. In: *Protein expression and purification* 152, S. 146–154. DOI: 10.1016/j.pep.2018.07.008.
- Jeacock, Kiani; Chappard, Alexandre; Gallagher, Kelly J.; Mackay, C. Logan; Kilgour, David P. A.; Horrocks, Mathew H. et al. (2023): Determining the Location of the α -Synuclein Dimer Interface Using Native Top-Down Fragmentation and Isotope Depletion-Mass Spectrometry. In: *Journal of the American Society for Mass Spectrometry* 34 (5), S. 847–856. DOI: 10.1021/jasms.2c00339.
- Wang, Wei; Perovic, Iva; Chittuluru, Johnathan; Kaganovich, Alice; Nguyen, Linh T. T.; Liao, Jingling et al. (2011): A soluble α -synuclein construct forms a dynamic tetramer. In: *Proceedings of the National Academy of Sciences of the United States of America* 108 (43), S. 17797–17802. DOI: 10.1073/pnas.1113260108.

Referee #2 (Comments on Novelty/Model System for Author):

I believe that this study is of great importance and translatability. First, the physiological role of alpha-synuclein tetramers is not yet fully understood. Second, we really need usable biomarkers of alpha-synuclein pathology in the blood, both for diagnostic purposes and to monitor the efficacy of therapies that counteract the disease.

Referee #2 (Remarks for Author):

I believe that this study is of great importance and translatability. First, the physiological role of alpha-synuclein tetramers is not yet fully understood. Second, we really need usable biomarkers of alpha-synuclein pathology in the blood, both for diagnostic purposes and to monitor the efficacy of therapies that counteract the disease. I do propose just minor revisions:

The thank the reviewer for her/his enthusiastic evaluation of our work.

as a general suggestion for introduction it can be better and more clearly stated that total soluble alpha-synuclein reflects three main states: it can be moderately decreased due to synucleinopathy, increased due to synaptic damage, and increased due to blood contamination.

The introduction has been revised accordingly (line 141-150, line 157).

Therefore, it is very ineffective as a biomarker of a single disease process. Seed amplification assays have been successfully applied to detect misfolded alpha-synuclein, but there is only one study conducted on blood, which many groups are trying to replicate without success. In addition, SAAs are very delicate techniques, very unstable, and few groups perform them successfully on a routine basis.

line 144 Serum albumin is an interactor of alpha-synuclein (PMID: 31034772), but lipoproteins might interact more with the aggregation kinetics of SAAs, making them inapplicable in clean plasma/serum (PMID: 37005644). The alpha-synuclein/liporothein interaction is already relevant in CSF and even more so in blood, where lipoproteins are about 30 times more concentrated. In fact, the only published positive result on SAA in serum was obtained by immunoprecipitating alpha-synuclein before applying SAA.

We agree that SAAs are still challenging, especially in blood. The lipoprotein interaction has not been mentioned before and is now added to introduction (line 187, line 191-195).

I hope that this line of research will be continued and that the ratio of tetrameric to monomeric alpha-synuclein will be tested on larger cohorts. It will be necessary, however, for the experimental procedures for measurement to be further standardized so that they can be easily replicated by other groups.

25th Apr 2024

Dear Prof. Bartels,

Thank you for the submission of your revised manuscript to EMBO Molecular Medicine. We have now received the enclosed reports from the referees that were asked to re-assess it. As you will see the reviewers are now globally supportive and I am pleased to inform you that we will be able to accept your manuscript pending the following final amendments:

- 1) Mass spectrometry data deposition: Please deposit all mass spectrometry raw data to a suitable community repository (ProteomeXchange member repository for protein mass spectrometry data) and provide the accession codes in the Data Availability statement. Please also ensure that the deposited datasets are publicly accessible.
- 1) Please rename "Financial Disclosure/Conflict of Interest" to "Disclosure and competing interests statement". We updated our journal's competing interests policy in January 2022 and request authors to consider both actual and perceived competing interests. Please review the policy <https://www.embopress.org/competing-interests> and update your competing interests if necessary.
- 2) In the Materials and Methods, please take care of the following:
 - Mass spectrometry: Please edit your manuscript to include the details of the mass spectrometry in the Methods as follows: state the total number of samples analyzed, numbers and types of controls, number of technical and/or biological replicates (even if n=1), provide a detailed description of the database search parameters and acceptance criteria used for peptide identification, and describe and/or reference the statistical tests used for the proteomics data analyses
 - Human research participants: Please state, in the section where you declare that the study was authorized, that the experiments conformed to the principles set out in the WMA Declaration of Helsinki and the Department of Health and Human Services Belmont Report. Please note that this is a separate statement from the specific ethics committee approval and informed consent.
- 3) Please place individual sections of the manuscript in the following order: Title page - Abstract & Keywords - Introduction - Results - Discussion - Materials & Methods - Data Availability - Acknowledgements - Disclosure and Competing Interests Statement - The Paper Explained - For More Information - References - Figure Legends - Expanded View Figure Legends.
- 4) For the figures and figure legends, please take care of the following:
 - Please make sure to update the callouts of all figures in the main manuscript text (currently figure callouts are missing for Fig. 2B
 - Please note that a separate 'Data Information' section is required in the legends of figures 3a-b.
 - Please note that information related to n is missing in the legends of figure 2b; EV 2a-c; EV 3a-b.
 - Please note that the error bars are not defined in the legends of figures EV 2a-c; EV 3a-b.
- 5) Tables: Please remove the legend for Table EV1 from the main manuscript text and include it in the Table EV1 file.
- 6) Funding: Please note that funding information should be given in the "Acknowledgements" section (not in its own separate section). Please ensure that all funding sources are entered into the manuscript submission system i.e. please add Chan Zuckerberg Collaborative Paire Initiative Phase 2, UK Dementia Research Institute (DRI), which receives its funding from DRI Ltd., the UK Medical Research Council and Alzheimer's Society, and Alzheimer's Research UK. Please be sure to enter the full name of the funding institute and to provide project numbers where available.
- 7) EMM does not have a separate section for abbreviations in Articles. Please explain acronyms/abbreviations in the main text and remove the Abbreviations section.
- 8) Please check your synopsis text and image before submission with your revised manuscript. Please be aware that in the proof stage minor corrections only are allowed (e.g., typos).
- 9) Source Data: Please upload the Source Data as a single source data file (zipped) per figure, with the panels clearly visible in the folder structure. Currently it is not clear which panel each Source Data figure is associated with and what they represent, so we would suggest renaming the files such that it is clear for the reader. Table EV2 should not be an Expanded View Table, but rather should be included in the Source Data folders for Figures 2 and 3. Please remove the legend for this table from the main manuscript and included as a separate tab in the Source Data file. Please also ensure that the callout for Table EV2 is removed from the main text.
- 10) For more information: This space should be used to list relevant web links for further consultation by our readers. Could you identify some relevant ones and provide such information as well? Some examples are patient associations, relevant databases, OMIM/proteins/genes links, author's websites, etc...
- 11) As part of the EMBO Publications transparent editorial process initiative (see our policy here: https://www.embopress.org/transparent-process#Review_Process), EMBO Molecular Medicine will publish online a Peer Review File (PRF) to accompany accepted manuscripts. This file will be published in conjunction with your paper and will include the anonymous referee reports, your point-by-point response and all pertinent correspondence relating to the manuscript. Let us know whether you agree with the publication of the PRF and as here, if you want to remove or not any figures from it prior to publication. Please note that the Authors checklist will be published at the end of the PRF.
- 12) Please provide a point-by-point letter INCLUDING my comments as well as the reviewer's reports and your detailed responses (as Word file).

I look forward to reading a new revised version of your manuscript as soon as possible.

Yours sincerely,

Poonam Bheda

Poonam Bheda, PhD
Scientific Editor
EMBO Molecular Medicine

***** Reviewer's comments *****

Referee #1 (Remarks for Author):

I want to thank the authors for their timely rebuttals.

I am satisfied with the revision and the detailed explanation of how the bands were quantified to minimize the effect of the smears.

Referee #2 (Comments on Novelty/Model System for Author):

I already appreciated the authors' work. The revised version, also thanks to the comments of other reviewers, further improved the scientific quality of the manuscript. The only limitation I clearly see is the sample size but it is still an acceptable limit for a pilot study.

Referee #2 (Remarks for Author):

I think the methodological rigor of the work has been further improved. The opinion of this reviewer is that the work is now suitable for publication.

The authors addressed the minor editorial issues.

17th May 2024

Dear Prof. Bartels,

We are pleased to inform you that your manuscript is accepted for publication and is now being sent to our publisher to be included in the next available issue of EMBO Molecular Medicine.

Yours sincerely,

Poonam Bheda, PhD
Scientific Editor
EMBO Molecular Medicine
